



# African biomes are most sensitive to changes in $CO_2$ under recent and near-future $CO_2$ conditions

Simon Scheiter[1], Glenn R. Moncrieff[2], Mirjam Pfeiffer[1], and Steven I. Higgins[3]

[1]Senckenberg Biodiversity and Climate Research Centre (SBiK-F), Senckenberganlage 25, 60325 Frankfurt am Main, Germany
[2]Fynbos Node, South African Environmental Observation Network, Claremont 7735, South Africa
[3]Chair of Plant Ecology, University of Bayreuth, Universitätsstraße 30, 95440 Bayreuth, Germany

**Correspondence:** Simon Scheiter (simon.scheiter@senckenberg.de)

**Abstract.** Current rates of climate and atmospheric change are likely higher than during the last millions of years. Even higher rates of change are projected in CMIP5 climate model ensemble runs for some RCP scenarios. Mismatches between the speed of ecological processes such as physiological adaptation, demographic shifts or migration, and the speed of changes in environmental conditions imply lags between the transient vegetation state and the vegetation state expected under prevailing

environmental conditions. Here, we used a dynamic vegetation model, the aDGVM, to study lags between transient and committed vegetation in Africa under changing atmospheric $CO_2$ mixing ratio. We hypothesized that lag size increases with more rapidly changing $CO_2$ mixing ratio as opposed to slower changes in $CO_2$, and that disturbance by fire further increases these lags. Our model results confirm these hypotheses, revealing lags between vegetation state and environmental conditions and enhanced lags in fire-driven systems. Biome states, carbon stored in vegetation and tree cover in Africa are most sensitive to

changes in the $CO_2$ mixing ratio under recent and near-future levels, between approximately 300 and 500 ppm. These results have important implications for vegetation modellers as well as for management and policy making. Lag effects implicate that vegetation will undergo substantial changes in distribution patterns, structure and carbon sequestration even if emissions of fossils fuels and other greenhouse gasses are reduced and the climate system stabilizes. We conclude that modelers need to account for lags in models and data used for model testing and that policy makers need to consider lagged responses and

committed changes in the biosphere when developing adaptation and mitigation strategies.

## 1 Introduction

Climate and the composition of the atmosphere have been subject to substantial changes during Earth history (Beerling and Royer, 2011). For instance, paleo-records indicate that the expansion of forest vegetation during the Devonian dramatically

reduced the atmospheric $CO_2$ mixing ratio (Le Hir et al., 2011) and Milankovitch cycles cause periodic changes in the climate system and the atmosphere on millennial time scales (Milankovic, 1941; Hays et al., 1976). In addition to natural variability,





anthropogenic emissions of $CO_2$ and other green house gasses have contributed to global warming during the last decades. The 5th assessment report of the Intergovernmental Panel of Climate Change (IPCC) indicates further changes of the climate system in the future (IPCC, 2013, 2014a, b). Since the pre-industrial era, $CO_2$ increased from approximately 280 ppm to a current

value of approximately 400 ppm, and the most pessimistic IPCC RCP 8.5 climate change scenario projects $CO_2$ increases to approximately 950 ppm by 2100. Proxy data suggest that such $CO_2$ levels have not occurred since the Eocene/early Oligocene, more than 30 Myr ago (Beerling and Royer, 2011). Current carbon emission rates are unprecedented and higher than during the Paleocene-Eocene Thermal Maximum (PETM), a period with high carbon emissions some 56 million years ago (Zeebe et al., 2016).

The $CO_2$ increase projected in the IPCC RCP 8.5 scenario corresponds to an average increase of more than 6 ppm per year until the end of the century. In comparison, an increase from approximately 190 ppm during the last glacial maximum to a pre-industrial value of 280 ppm during 26,000 years corresponds to an average rate of $3.5 \times 10^{-3}$ ppm per year (Barnola et al., 1987). Within this period, Monnin et al. (2001) report peak rates of $2.7 \times 10^{-2}$ ppm per year in a 300 year period at 13.8ky B.P. A decrease from approximately 900 ppm to 300 ppm during the Oligocene took approximately 10 million years (Beerling and

Royer, 2011), which translates into an average rate of $-6 \times 10^{-5}$ ppm per year. As the current rates of $CO_2$ change are likely unprecedented, no proxy analogues exist to deduce vegetation responses to the ongoing atmospheric and climatic changes (Prentice et al., 1993; Foster et al., 2017). Due to the coarse temporal resolution of many paleo-records, it is, however, still challenging to calculate rates at decadal or even finer temporal resolution for a direct comparisons of past, present and future rates of change (but see Zeebe et al., 2016).

Environmental conditions such as $CO_2$, precipitation, temperature or soil properties influence plant ecophysiological processes that ultimately drive plant growth, demographic rates, competitive hierarchies, community assembly and biogeographic patterns across the Earth's land surface. These processes are sensitive to both the actual values and to variation of environmental drivers. Different ecological processes operate at various temporal and spatial scales (Penuelas et al., 2013) and determine how ecosystems respond to environmental change. On short time scales (days to months), plasticity allows photosynthesis (Gun-

derson et al., 2010), carbon allocation and other processes to adapt to changes in environmental conditions (Penuelas et al., 2013). At intermediate time scales (years to decades), vegetation is influenced by demographic rates, succession, dispersal, migration and community assembly (Penuelas et al., 2013). At long time scales (centuries and longer), evolutionary processes allow plants to adapt to changing environments and speciation and extinction modify the species pool.

   A mismatch between rates of change in the environmental forcing and ecological responses of vegetation implies that veg-

etation is not in an equilibrium state with the environment, that is a state where averages of key ecosystem functions such as carbon and water fluxes or vegetation structure remain in constant if averages of environmental drivers remain in constant. Rather, forcing lags emerge where the transient vegetation state lags behind rates of change of environmental drivers (Bertrand et al., 2016). Quantifying these lags is crucial for our understanding of ecosystem dynamics because they imply that the observed vegetation state does not fully reflect prevailing environmental conditions and that ecosystems are committed to further

changes even if environmental conditions stabilize (Jones et al., 2009; Port et al., 2012). We expect that the size of the forcing lag between equilibrium and transient vegetation states will be influenced by actual values of environmental conditions, the





rate at which they change, and by the plant community composition and community-specific ecological processes. Vegetation might have been closer to equilibrium with the environment, with smaller lags, in the past when rates of environmental changes were low, while it is committed to large changes under current and future, rapidly changing climate.

Disturbances such as fire, drought, heat waves or herbivory rapidly modify vegetation states and thereby create disturbance lags, i.e., deviations from the committed vegetation state purely defined by environmental conditions in the absence of disturbance. Lag size is related to the intensity of the respective disturbance. Abrupt and repeated disturbances imply that vegetation oscillates between different successional stages. In such a situation, vegetation may never reach the final successional stage but it may be in a dynamic equilibrium state. The ecological resilience of an ecosystem (Holling, 1973; Walker et al., 2004) influ-

ences if the system can return into a pre-disturbed state or if it tips into an alternative vegetation state (Scheffer et al., 2001; van Nes and Scheffer, 2007; Veraart et al., 2012). Savannas exemplify an ecosystem type that is strongly influenced by and often reliant on disturbances (Scheiter and Higgins, 2007), and subject to both disturbance and forcing lags. In savannas, fire reduces woody biomass to the benefit of grasses. However, once fire disturbance is removed, fire-driven savannas are committed to transition to higher tree cover (Sankaran et al., 2004; Higgins et al., 2007; Higgins and Scheiter, 2012). It has been argued

that alternative vegetation states are possible in areas currently covered by savannas, because depending on their history and fire activity, they can adopt an open savanna state or a closed forest state (Higgins and Scheiter, 2012; Moncrieff et al., 2014). We expect that in areas that allow both savanna and forest states, fire amplifies forcing lags. In such systems, environmental forcings need to cross a tipping point such that vegetation shifts from one ecosystem state into an alternative state (Scheffer et al., 2001).

Deciphering and quantifying lags between transient and equilibrium vegetation states is highly relevant for understanding biogeographic patterns and associated biogeochemical fluxes as well as for conservation and management. The importance of transient states and forcing lags was already highlighted in the 1980s, but often in the context of paleo-ecological studies (Davis and Botkin, 1985; Davis, 1986; Webb III, 1986). For instance, Davis and Botkin (1985) found lagged responses of various species in response to cooling using the JABOWA vegetation model. Changes in the dominance of species were only

visible 50 years after cooling. Several empirical studies quantified lag effects, for example in forests (Bertrand et al., 2011, 2016; Liang et al., 2018), bird and butterfly communities (Devictor et al., 2012; Menendez et al., 2006), or in tropical forests at the global scale (Zeng et al., 2013). Most of these studies investigated lags with respect to recent temperature changes or in mountain areas with steep temperature gradients. Lag effects were also identified in response to drought (Anderegg et al., 2015). More recently, lag effects received more attention in the context of future climate change. It has been argued that lag

effects need to be taken into account when we aim at forecasting future changes in the biosphere and at developing management or mitigation strategies (Svenning and Sandel, 2013; Bertrand et al., 2016). Lag effects imply that vegetation features such as carbon stocks or tree cover are committed to changes that will be ongoing even if anthropogenic emissions of greenhouse gasses level off and the climate system stabilizes (Jones et al., 2009; Port et al., 2012; Huntingford et al., 2013; Pugh et al., 2018). Yet, previous studies did not consider a large $CO_2$ gradient ranging from pre-industrial to future levels, but rather post-2100

conditions.



In this study, we use aDGVM, a complex dynamic vegetation model developed for tropical grass-tree ecosystems (Scheiter and Higgins, 2009) to investigate how slow and fast changes in atmospheric $CO_2$ and fire regimes influence transient and equilibrium distributions of grasslands, savannas and forests in Africa, as well as associated biomass and tree cover. aDGVM is an appropriate modeling tool in this context because it explicitly simulates the rate at which vegetation changes based on underlying ecophysiological processes and underlying environmental conditions. It allows us to simulate both the equilibrium vegetation state for given $CO_2$ mixing ratios, and transient vegetation dynamics, succession and adaptation of photosynthesis, evapotranspiration, carbon allocation and phenology. We focus on atmospheric $CO_2$ because it is a main driver of plant growth and both empirical and modeling studies have shown substantial impacts on vegetation growth (Scheiter and Higgins, 2009; Buitenwerf et al., 2012; Higgins and Scheiter, 2012; Donohue et al., 2013; Hickler et al., 2015). In addition, the $CO_2$ mixing ratio is almost similar at the global scale while other key drivers of plant growth such as rainfall and temperature are highly variable spatially (i.e. between different regions of the world), temporally (i.e. inter- and intra-annual variability) and between different climate models within the CMIP5 ensemble.

We test the following predictions: (1) vegetation is typically not in equilibrium with the environment, more specifically with atmospheric $CO_2$, and forcing lags occur; (2) the size of the forcing lag is influenced by the rate of change of $CO_2$; (3) disturbance lags due to fire amplify forcing lags caused by $CO_2$ change such that biomes with high fire activity will take longer to keep pace with environmental changes; (4) the sensitivity of vegetation to changes in the atmospheric $CO_2$ mixing ratio is sensitive to the absolute value of the $CO_2$ mixing ratio. We explore the consequences of these predictions for projections of climate change impacts on African vegetation under rates of $CO_2$ change as predicted in RCP 2.6, 4.5, 6.0 and 8.5, examining the difference between transient and equilibrium vegetation states as the $CO_2$ mixing ratio change.

## 2 Methods

### 2.1 Model description

We used the aDGVM (adaptive Dynamic Global Vegetation Model, Scheiter and Higgins, 2009), a dynamic vegetation model for tropical grass-tree systems. The aDGVM integrates plant physiological processes generally used in dynamic global vegetation models (DGVMs, Prentice et al., 2007) with processes that allow plants to dynamically adjust leaf phenology and carbon allocation to environmental conditions. The aDGVM is individual-based and simulates state variables such as biomass, height and photosynthetic rates of individual plants. This approach allows to model how herbivores (Scheiter and Higgins, 2012), fire (Scheiter and Higgins, 2009) and land use (Scheiter and Savadogo, 2016; Scheiter et al., 2019) impact individual plants as a function of plant traits. Grasses are simulated by two super-individuals, representing grasses beneath or between tree canopies.

The aDGVM simulates four plant types (Scheiter et al., 2012): $C_3$ grasses, $C_4$ grasses, fire-sensitive forest trees and fire-tolerant savanna trees. The differences between $C_3$ and $C_4$ grasses are mainly based on physiological differences between $C_3$ and $C_4$ photosynthesis. Savanna and forest tree types differ in fire- and shade-tolerance (Bond and Midgley, 2001; Ratnam et al., 2011). The forest tree type is implemented to be shade-tolerant but fire-intolerant whereas the savanna tree type is shade-





intolerant but fire-tolerant. Hence, forest trees dominate in closed ecosystems and in the absence of fire, whereas savanna trees dominate in fire-driven and more open ecosystems.

In the aDGVM, fire intensity is modeled as a function of fuel loads, fuel moisture and wind speed (Higgins et al., 2008). Fire spreads when (1) the fire intensity exceeds a threshold value of 300 kJ/m/s, (2) a uniformly distributed random number exceeds the daily fire ignition probability $p_{fire}$ (1%), and (3) an ignition takes place. Ignition sequences, which indicate days when ignitions take place, are randomly generated. This fire model ensures that fire regimes are influenced by fuel biomass and climate. However, fire ignitions and the ignition probability are not linked to anthropogenic ignitions or environmental

conditions such as the occurrence of lightning. Fire removes aboveground grass biomass, whereas the response of trees to fire is a function of tree height and fire intensity (topkill effect, Higgins et al., 2000). Seedlings and juveniles in the flame zone are damaged by each fire while tall trees with tree crowns above the flame zone are largely fire-resistant and only damaged by intense fires. Grasses and topkilled trees can regrow from root reserves after fire (Bond and Midgley, 2001). Fire influences tree mortality indirectly due to its negative effect on the carbon balance. In the aDGVM, a negative carbon balance increases

the probability of mortality.

    The performance of the aDGVM was evaluated in previous studies. Scheiter and Higgins (2009) and Scheiter et al. (2012) show that the aDGVM successfully simulates the distribution of major vegetation formations in Africa in good agreement with observations. Scheiter and Higgins (2009) show that the aDGVM can simulate biomass dynamics observed in a long term fire manipulation experiment in the Kruger National Park (Experimental Burn Plots, Higgins et al., 2007). In Scheiter

and Savadogo (2016) we showed that a slightly adjusted model version can reproduce grass biomass and tree basal area under different grazing, harvesting and fire treatments in Burkina Faso.

## 2.2   Biome classification

We classify vegetation into biome types using the classification scheme presented in Scheiter et al. (2012) and used in previous aDGVM studies. When grass biomass in a simulated grid cell is less than 0.5 t/h and total tree cover is less than 10%, vegetation

is classified as desert or barren. When tree cover is less than 10% and grass biomass exceeds 0.5 t/ha, vegetation is, depending on the ratio of $C_3$ to $C_4$ grasses, classified as $C_3$ or $C_4$ grassland. At intermediate tree cover between 10% and 80%, the ratio of $C_3$ to $C_4$ grass biomass and the cover of savanna and forest trees are used for classification. Vegetation is classified as a woodland if forest tree cover exceeds savanna tree cover, whereas vegetation is classified as savanna if savanna tree cover exceeds forest tree cover. We distinguish between $C_3$ savanna and $C_4$ savanna, depending on the ratio of $C_3$ to $C_4$ grasses.

Vegetation is classified as forest when tree cover exceeds 80%, irrespective of tree type and grass biomass.





## 2.3 Equilibrium conditions for aDGVM

We assume that an aDGVM state variable $V_i$ at time $i$ (see next paragraph for state variables used in the analysis) is in equilibrium if

$$\sum_{i=Y-l+1}^{Y} |V_i - V_{i-1}| < \epsilon, \tag{1}$$

where $Y$ is the current year of the simulation, $l$ is the number of years used for the calculation of equilibrium conditions (we use $l = 30$) and $\epsilon$ is a threshold defining the narrowness of the equilibrium (we use $\epsilon = 0.001$). Trial simulations show that these values allow vegetation to reach equilibrium within feasible model simulation runtime. Using different threshold values changed the time required to reach the equilibrium state but did not change our basic results. Choosing $\epsilon$ too small will identify model stochasticity as deviation from equilibrium, whereas choosing $\epsilon$ too large will fail to correctly identify the onset of

equilibrium conditions. Systematic sensitivity analyses for $\epsilon$ and $l$ were not conducted.

We used four modeled state variables $V$ to characterize equilibrium states: savanna tree cover, forest tree cover, aboveground tree biomass and $C_3$:$C_4$ grass ratio. We assume that the model is in equilibrium when all four variables fulfill eq. (1) simultaneously and we record the first year when the model is in equilibrium, $Y_e$. It is possible that one or several variables leave the equilibrium state again after year $Y_e$, and that the condition in eq. (1) is no longer met for these variables. This can be for

example due to stochasticity in rainfall or due to fire. However, such situations are not considered in our analysis.

## 2.4 Simulation experiments

All simulations were conducted for Africa at $2°$ spatial resolution. In all simulation scenarios, we initialized aDGVM with 100 small trees of both types and two super-individuals representing grasses under and between tree crowns. All simulation scenarios in this study manipulated only $CO_2$ whereas other climate variables such as precipitation or temperature were kept

constant with monthly climatology provided by CRU (Climatic Research Unit, New et al., 2002) for the reference period between 1961 and 1990. This model design allows us to study $CO_2$ effects in isolation and it avoids interactive effects of several forcing variables on the system state.

To test the first prediction, i.e., that vegetation is not in equilibrium with the environment, we simulated (1) the equilibrium vegetation state for different $CO_2$ mixing ratios, and (2) transient vegetation dynamics with increasing and decreasing $CO_2$

mixing ratio. For simulations of the equilibrium state, we set $CO_2$ to 100 ppm, 150 ppm, ..., 1000 ppm. For each $CO_2$ level we ran the model until an equilibrium vegetation state was reached in year $Y_e$ (eq. 1). Equilibrium conditions were derived for each simulated $2°$ grid cell separately. For each grid cell and each $CO_2$ level, we classified vegetation in year $Y_e$ into biome types. We calculated the fractional area covered by different biome types in equilibrium and used the 'loess' smoother in R (R Core Team, 2018) to obtain continuous response curves of fractional cover of biomes with respect to $CO_2$. We further created

maps of the spatial patterns of $Y_e$.

For simulations of the transient vegetation state, aDGVM was initialized at low (100 ppm) or high (1000 ppm) $CO_2$ mixing ratio. In each grid cell, simulations were conducted until vegetation fulfilled the equilibrium condition defined in eq. (1). We





then increased or decreased $CO_2$ linearly to 1000 ppm or 100 ppm by 3.5 ppm per year or 0.9 ppm per year (see second prediction for justification of these rates). We used linear $CO_2$ changes between a minimum and a maximum $CO_2$ mixing ratio

because linear changes in forcing variables allow to identify non-linear and tipping point behaviour in the vegetation state (Scheffer et al., 2001). Once $CO_2$ reached 1000 ppm or 100 ppm, respectively, simulations were continued until vegetation re-established the equilibrium state according to eq. (1) and the duration was tracked. For each grid cell and each simulation year, we classified vegetation into biomes and calculated fractional cover of each biome type. The difference between the vegetation state when first reaching a target $CO_2$ mixing ratio in transient runs and the equilibrium vegetation state at the target $CO_2$

mixing ratio is an indicator of the lag size between environmental forcing and vegetation. We used the proportion of Africa covered by different biome types, woody biomass and tree cover as proxies of lag size.

To test the second prediction, i.e., that the difference between transient and equilibrium vegetation is influenced by the rate of change of environmental forcings, we conducted transient model runs where $CO_2$ changed at two different rates. Specifically, $CO_2$ mixing ratios were changed by 3.5 ppm per year or 0.9 ppm per year to represent current and past rates of change.

The higher rate represents the average $CO_2$ increase in the RCP 6.0 scenario, where $CO_2$ increases from current values to approximately 700 ppm in 2100. In the simulations, 3.5 ppm per year implies that $CO_2$ changes from 100 ppm to 1000 ppm (or vice versa) within 230 years. The 0.9 ppm per year rate of $CO_2$ change implies a change from 100 ppm to 1000 ppm (or vice versa) within approximately 900 years. This rate overestimates rates of $CO_2$ change at paleo-ecological time scales. It was selected such that it differs substantially from the higher rate but that model run time is still feasible.

To test the third prediction, i.e., that fire amplifies lag effects, we conducted all simulations described in the previous paragraphs with fire switched on or off.

To test the fourth prediction, i.e., that sensitivity of vegetation to changes in $CO_2$ is influenced by the $CO_2$ mixing ratio, we calculated the sensitivity of modeled state variables in relation to changes in $CO_2$,

$$\delta_V(C) = \frac{|V(C) - V(C + \Delta C)|}{\Delta C} \qquad (2)$$

Here, $V(C)$ is an aDGVM state variable at $CO_2$ mixing ratio $C$, and $\Delta C$ is the increment of the $CO_2$ mixing ratio used to calculate sensitivity. Sensitivity was calculated for the entire gradient considered in the study (i.e. 100 ppm to 1000 ppm) and for all scenarios (i.e. equilibrium and transient, with and without fire). To filter out variability of simulated variables due to model stochasticity and to account for different rates of change of the $CO_2$ mixing ratio in different scenarios, we used the 'loess' function in R (R Core Team, 2018) for smoothing. The smoothed curves were used for calculations of sensitivity.

To explore if vegetation is currently in equilibrium with the atmospheric $CO_2$ mixing ratio or committed to further change until 2100, we conducted simulations with $CO_2$ from the RCP 2.6, RCP 4.5, RCP 6.0 and RCP 8.5 scenarios between 1765 and 2100 (Meinshausen et al., 2011). Climate conditions were kept constant with monthly climatology provided by New et al. (2002) to be able to compare simulations for different RCP scenarios to equilibrium simulations described in previous paragraphs. We compare the simulated vegetation state in these transient runs to the equilibrium vegetation state at selected

$CO_2$ mixing ratios to quantify lags in carbon and tree cover.





We conducted simulations at one selected savanna study site in South Africa (26°S, 28°E) to illustrate how different state variables simulated by aDGVM respond to $CO_2$ increases between 100 ppm and 1000 ppm at a rate of 3.5 ppm per year. To account for stochastic effects in aDGVM we conducted 200 replicate simulation runs. Simulations were conducted with fire. We analyzed leaf level photosynthetic rates, tree numbers, maximum tree height, mean tree height, forest tree cover and savanna

tree cover, averaged for all replicate runs. We plotted time series of these variables both in their native units and normalized between 0 and 1 using minimum and maximum values of the variables to be able to track the temporal lags in these variables.

## 3 Results

### 3.1 Equilibrium vegetation state

Equilibrium simulations for fixed $CO_2$ mixing ratios show that the cover of $C_4$-dominated vegetation ($C_4$ grasslands and

savannas) in Africa decreases with increasing $CO_2$, whereas the area covered by $C_3$-dominated woody vegetation (woodlands and forests) increases (Fig. 1, 2). This general pattern is simulated both in the presence and the absence of fire. Fire increases the cover of more open $C_4$-dominated vegetation states at the expense of $C_3$ woody vegetation states. This result indicates that large areas in Africa can be covered by $C_4$- or $C_3$-dominated vegetation if fire is present or absent. The proportions of the area where either $C_4$- or $C_3$-dominated vegetation is possible peaks at low $CO_2$ (approximately 200 ppm) and decreases at higher

$CO_2$ mixing ratios. The area covered by $C_3$-dominated woody vegetation is maximized at 1000 ppm and saturates at 69% in the absence of fire and at 61% in the presence of fire. The area covered by $C_4$-dominated vegetation peaks at 49% at a low $CO_2$ mixing ratio in the presence of fire and at 16% in the absence of fire. The area covered by deserts decreases from 46% to 24% as $CO_2$ increases and these areas are replaced by grasslands, savannas and woodlands (Fig. 1, 2). Areas covered by $C_3$ grasslands and $C_3$ savannas increase as $CO_2$ increases, but even at 1000ppm, coverage is less than 10%, irrespective of the

presence or absence of fire (Fig. A1).

The time until vegetation reaches an equilibrium state varies substantially in different biomes and for different $CO_2$ mixing ratios. Times are longest in more open ecosystems, that is in grasslands, woodlands and savannas (Fig. 3). In most biomes, times tend to increase with $CO_2$. Times are shortest in forests. The duration is generally longer in the presence of fire than in the absence of fire.

### 3.2 Transient vegetation state and forcing lags

When vegetation is initialized at a $CO_2$ mixing ratio of 100 ppm or 1000 ppm and $CO_2$ increases or decreases progressively in transient simulations, the area covered by different biome types at a given $CO_2$ deviates considerably from the cover in equilibrium simulations (Fig. 4). This pattern is consistent both in simulations with and without fire. Deviance indicates that vegetation is generally not in equilibrium with the environment and that it lags behind the environmental forcing. At low and

increasing $CO_2$, the area covered by grasslands and savanna increases steeply and overshoots the initial cover at 100 ppm, mainly because grasslands invade into deserts. As $CO_2$ increases, the areas covered by $C_3$- and $C_4$-dominated vegetation





approach the equilibrium state. Trees suppress grasses and eventually fire occurrence, such that forests ultimately manage to invade most of the vegetated area. The deviation between equilibrium and transient cover of $C_3$-dominated vegetation for slow changes of $CO_2$ in the presence of fire (Fig. 4d) is due to transitions from $C_4$ grasslands and $C_4$ savannas to $C_3$ grasslands and
$C_3$ savannas.

The aDGVM simulates similar lag effects when $CO_2$ decreases, indicating hysteresis, which is a signature of alternative ecosystem states. The change of vegetation cover in transient simulation runs is non-linear although the $CO_2$ forcing changes linearly. In summary, this result confirms our first prediction, i.e. transient vegetation states deviate from the equilibrium state and forcing lags occur.

The rate at which $CO_2$ increases or decreases has a strong impact on the size of the forcing lags (Fig. 4). Fast changes in $CO_2$ imply larger lags than slow changes in $CO_2$. Lags are larger at low and intermediate $CO_2$ mixing ratios and decrease at higher $CO_2$, irrespective of the rate of change of $CO_2$. This result verifies the second prediction.

### 3.3 Fire and disturbance lags

A comparison of simulations with and without fire shows that fire increases the lag between the $CO_2$ forcing and vegetation
(Fig. 4). In the scenario with rapidly changing $CO_2$ mixing ratio, the maximum lag size averaged for entire Africa for $C_4$-dominated biomes is 20% and 16.3% with and without fire, respectively, the maximum lag size for $C_3$-dominated woody biomes is 19.4% and 16% with and without fire, respectively. In scenarios with increasing $CO_2$, lag size is maximized between approximately 200ppm and 300ppm. When integrated along the entire $CO_2$ gradient, the mean lag size for $C_4$-dominated biomes is 11.8% and 6.7% with and without fire, respectively, the mean lag size for $C_3$-dominated biomes is 11.9% and
6.4% with and without fire, respectively. These patterns are similar in simulations with slowly changing $CO_2$ mixing ratio, but percentages are systematically lower. The time to reach equilibrium is longer in simulations with fire than in simulation without fire (Fig. 3), in particular in fire-driven biome types. Times are similar in forests with dense tree canopy, where aDGVM does not simulate fire.

When $CO_2$ is held constant after the transient phase, vegetation converges towards the equilibrium state. In simulations
without fire, times to reach equilibrium were similar in equilibrium simulations and in transient simulations with both slow or fast increases or decreases of $CO_2$ (Figs. 5, 6). In simulations with fire, we generally observed longer times than in simulations without fire, particularly in grassland and savanna areas and for decreasing $CO_2$ (Fig. 6). In simulations with decreasing $CO_2$ and fire included, times to reach equilibrium in grasslands and savannas were considerably longer in transient simulations than in equilibrium simulations (Fig. 6). These findings support our third prediction that disturbance lags amplify forcing lags and
are particularly relevant in fire-driven systems.

### 3.4 Carbon and tree cover debt

Forcing lags and disturbance lags imply carbon and tree cover debt (for increasing $CO_2$, Fig. 7) or surplus (for decreasing $CO_2$, Fig. A2). Debt means that at a given $CO_2$ mixing ratio, tree cover and carbon stocks are lower in transient simulations than in equilibrium simulations at the same $CO_2$ level. Accordingly, surplus means that tree cover and carbon are higher in





transient simulations than in equilibrium simulations. Where debts and surpluses occur, carbon and tree cover are committed to further changes, even if environmental forcings stabilize, unless tipping point behavior inhibits vegetation change or allows rapid vegetation changes that compensate debt or surplus.

Carbon debt increases over the entire $CO_2$ gradient. At current $CO_2$ levels of 400 ppm, carbon debt for Africa is between -6.3 and -13.6 PgC for different scenarios and it increases to values between -24.8 and -39.9 PgC for 1000 ppm. At high $CO_2$, 285  debt is higher for simulations without fire than for simulations with fire due to the combined effects of forcing and disturbance lags.

Tree cover debt in the presence of fire peaks at values around -15% between 300 ppm and 400 ppm, i.e. at current $CO_2$ levels. Debt decreases at higher $CO_2$ mixing ratios to values between -5.8% and -10%, depending on the scenario. Debt is generally higher in presence of fire and when $CO_2$ changes rapidly. The maximum deviance between transient and equilibrium 290  state (maximum debt and surplus) varies spatially with higher deviance in savannas and woodlands surrounding the central African forest and in the presence of fire.

### 3.5 Sensitivity of carbon and tree cover change

The sensitivity of the fractional cover of different biome types to changes in $CO_2$ is influenced by the actual values of $CO_2$ (Fig. 8a, b). In equilibrium simulations, the cover of $C_3$-dominated biomes is most sensitive at low $CO_2$ mixing ratios and 295  it decreases as $CO_2$ increases. In simulations with fire, sensitivity of $C_4$-dominated biomes is hump-shaped with a peak at ca. 380 ppm. In transient simulations, both $C_3$ and $C_4$-dominated biomes are most sensitive to changes in $CO_2$ mixing ratios between 200 and 600 ppm, depending on the specific scenario (Fig. 8c, d). Sensitivity is generally higher in the presence of fire than in the absence of fire. For rapidly changing $CO_2$, the peak is found at higher $CO_2$ mixing ratios than for slowly changing $CO_2$. These results support our fourth prediction.

### 3.6 Responses to RCP scenarios

Simulations for $CO_2$ mixing ratios following trajectories of different RCP scenarios indicate carbon debt (Fig. 9a) and tree cover debt (Fig. 9b) as the $CO_2$ mixing ratio increases, similar to the simulations with linear changes of $CO_2$ (Fig. 7). At the current $CO_2$ mixing ratio of approximately 400 ppm, aboveground tree carbon debt is between -8.9 PgC without fire and -16.5 PgC with fire. In the RCP 2.6 and 4.5 scenarios, carbon debt in Africa accumulates to peak values between -9.9 and 305  -18 PgC and between -12.9 and -22 PgC, respectively and then decreases because in these scenarios, $CO_2$ decreases (RCP 2.6) or saturates (RCP 4.5) at the middle of the century. In RCP 6.0, debt accumulates to values between -21.2 and -31 PgC, in RCP 8.5 it accumulates to values between -47.5 and -60 PgC until 2100. In contrast, tree cover debt peaks between ca. 300 and 350 ppm at values between -3.3 and -17% in the presence or absence of fire, respectively. Tree cover debt decreases as the $CO_2$ mixing ratio increases towards the end of the century, with a rate depending on the specific RCP scenario. Generally, both 310  carbon and tree cover debt are higher in the presence of fire than under fire suppression.



## 4 Discussion

Using a dynamic vegetation model, we show that vegetation is typically not in equilibrium with environmental conditions, and that transient vegetation deviates from the vegetation state expected for the prevailing environmental conditions. Vegetation dynamics lag behind changing environmental drivers due to forcing lags. The size of the forcing lag depends both on actual

environmental conditions and on the rate at which conditions change. Disturbance lags caused by fire can amplify forcing lags in areas where multiple vegetation states are possible such as savanna areas that also support forest. Our results indicate that vegetation in Africa is most sensitive to changes in the atmospheric $CO_2$ mixing ratio at current conditions. Hence, even if anthropogenic emissions of $CO_2$ and the accumulation of $CO_2$ in the atmosphere should level off in near future, ecosystems will still be committed to considerable changes.

Our model simulations are consistent with our previous findings that biome distributions over large areas of Africa are dependent on fire (Higgins and Scheiter, 2012), and that these distributions are contingent on historic vegetation states and likely to change under elevated $CO_2$ (Scheiter and Higgins, 2009; Higgins and Scheiter, 2012; Moncrieff et al., 2014). While we only considered transient vegetation dynamics in previous studies, we now show that these results also hold true for simulations with equilibrium conditions. In our transient simulations we further show that linear forcing in $CO_2$ can cause non-linear

responses in vegetation states, an indication of internal feedback loops and tipping point behaviour in the climate-fire-vegetation system (Scheffer et al., 2001). The potential for alternate biomes, dependent on fire, and hysteresis effects occur both in equilibrium simulations with fixed $CO_2$ and in transient simulations with variable $CO_2$ mixing ratio. These effects occur over the entire $CO_2$ gradient between 100 and 1000 ppm.

### 4.1 Understanding forcing, disturbance and successional lags

Lags between environmental conditions and vegetation states occur if environmental conditions change faster than vegetation can respond. In such a situation, transient vegetation states deviate from the vegetation states that one would expect if prevailing environmental conditions remained constant for a sufficiently long duration. The lag size is defined by the integrated effect of interacting processes including delayed responses in ecophysiology, demography, migration and succession, and by the different timescales on which these processes operate (Penuelas et al., 2013). Rates of change in environmental forcing and intensity

and frequency of disturbances further influence lag size. In aDGVM simulations we find a sequence of vegetation responses to changes in the $CO_2$ mixing ratio that operate at different temporal scales. When $CO_2$ increases, leaf level photosynthesis and respiration increase instantaneously following the ecophysiology models implemented in aDGVM (Farquhar et al., 1980; Collatz et al., 1991, 1992, Fig. 10). These adaptations imply higher carbon gain, water use efficiency and growth of individual trees in the growing season as well as higher reproduction rates, because in aDGVM, the amount of carbon allocated to repro-

duction is a function of peak carbon gain. Free air carbon enrichment (FACE) experiments and open top chamber experiments for elevated $CO_2$ indicate similar responses at the leaf level (Hickler et al., 2015; Kgope et al., 2010, Raubenheimer, Ripley, et al., unpublished).





In aDGVM, higher growth and reproduction rates of individual plants modify plant population dynamics and vegetation structure. The model simulates an increase in mean tree height after $CO_2$ starts increasing (Fig. 10). After a delay of approxi-

mately 70 years, trees can establish more successfully and tree number and savanna tree cover increase. Tree cover increases can be attributed to both increases in tree height, which in aDGVM implies an increase in a tree's crown area, and in tree number. Increases in maximum tree height lag behind mean tree height and tree numbers (Fig. 10). All population-level responses in the model are slower than leaf-level responses and lag behind ecophysiological adaptations.

Observing population-level responses to elevated $CO_2$ in reality is challenging. Historic data from field surveys (Stevens

et al., 2017; O'Connor et al., 2014) and remote sensing (Donohue et al., 2013; Skowno et al., 2017) indicate woody encroachment in many savanna areas. These changes were often attributed to historic increases in $CO_2$ (Midgley and Bond, 2015), but also to land-use activities such as over-grazing (Roques et al., 2001). Yet, the strength of $CO_2$ fertilization effects is debated (Körner et al., 2005) and free air carbon enrichment (FACE) experiments indicate complex responses of vegetation to elevated $CO_2$ at population level (Hickler et al., 2015). Nutrient limitations (Hickler et al., 2015) and effects of mycorrhizal

associations on nutrient economy (Terrer et al., 2016) may add to the complexity of these responses. FACE experiments in savannas and sub-tropical ecosystems are rare. An exception is OzFACE in Queensland, Australia (Stokes et al., 2005) which found increased growth rates for *Eucalyptus* and *Acacia* species. Previous studies showed that $CO_2$ fertilization effects are strong in aDGVM and largely compensate changes of other environmental conditions, particularly decreases in precipitation (Scheiter et al., 2015). In Scheiter et al. (2018) we show that simulated woody cover increases in the Limpopo Province, South

Africa, under current conditions broadly agree with observations from Stevens et al. (2017), indicating that aDGVM simulates plausible responses to climate change for historic and current conditions.

Disturbance lags in fire-driven savannas are created by a well-known feedback mechanism between fire and vegetation (Higgins and Scheiter, 2012; Hoffmann et al., 2012). Regular fire is a demographic bottleneck for tree establishment and traps trees in a juvenile state (Higgins et al., 2000). At the population scale, fire preserves a characteristic, open savanna vegetation

state (Scheiter and Higgins, 2009) and keeps vegetation from reaching an equilibrium vegetation state, typically woodland or forest with higher tree cover, expected under the prevailing environmental conditions. Yet, reduced fire activity (Hoffmann et al., 2012) or increased tree growth rates due to $CO_2$ fertilization (Bond and Midgley, 2000) allow more trees to escape the fire trap due to increased growth rates. As a consequence, the increasing tree cover starts to exclude grasses and curtails fire frequency due to reduced fine fuel loads. This dynamic feedback between vegetation and fire dynamics implies that rapid

transitions between savanna and forest states are possible.

Once fire is excluded at high $CO_2$ mixing ratio, successional lags delay the establishment of an equilibrium state. These lags are a direct consequence of disturbances and they emerge if plant community composition in the equilibrium state deviates from transient, post-disturbance community composition. The aDGVM simulates fire-tolerant but shade-intolerant savanna trees and fire-intolerant but shade-tolerant forest trees (Scheiter et al., 2012). At low $CO_2$ and intermediate rainfall, aDGVM simulates

a fire-driven savanna state with predominantly savanna trees. Reduced fire activity and transitions to woody plant-dominated habitats imply that savanna trees and grasses are out-competed and gradually replaced by forest trees (Fig. 10). Successional dynamics are slow and delay adaptation of community composition to elevated $CO_2$. Empirical studies support successional





lags in communities (Fauset et al., 2012; Esquivel-Muelbert et al., 2019), yet, these studies observed community responses to drought rather than $CO_2$.

We found that fire-driven savanna and grassland ecosystems take longer to reach equilibrium after the $CO_2$ forcing stabilizes than forests. Forests are faster to stabilize and to balance their carbon and tree cover debt. In our simulations, this behavior is driven by disturbance-related lags caused by fire. Fire generates a dynamic disequilibrium between climate and vegetation, and it prevents both savanna and forest trees to recruit, to transition into the adult state, and to develop a closed canopy. When tree cover exceeds a critical threshold, fire is suppressed and rapid canopy closure is possible. As fire rarely occurs in our simulated

forests, they lack the disturbance-related lag component and therefore reach equilibrium faster than fire-affected biome types. Moreover, forests in aDGVM represent the final stage in succession. Hence, they do not allow for further succession in contrast to grasslands and savannas, where savanna trees can invade grasslands, and be replaced by forest trees in later successional stages. aDGVM might underestimate lags in forest systems in contrast to alternative DGVMs or forest models that simulate a higher number of PFTs or species (e.g. Hickler et al., 2012), or that allow plant traits and community composition within a

forest system to adapt to changing environmental drivers (Scheiter et al., 2013).

### 4.2    Implications for adaptation, mitigation and policy

Lags between transient and equilibrium coverage of different vegetation types or biome types imply debt or surplus in tree cover (Jones et al., 2009), carbon storage, biogeochemical fluxes and community composition (Bertrand et al., 2016). These lags commit ecosystems to further changes even if the rate of climate change is reduced and the climate system converges

towards an equilibrium state (Jones et al., 2009; Port et al., 2012; Pugh et al., 2018). This finding has important implications for the development of adaptation and mitigation strategies for climate change.

First, it indicates that such strategies cannot be developed purely based on observed contemporary transient states when attempting to mitigate further changing of climatic drivers, as is currently often the case. There is an urgent need to understand equilibrium vegetation states, committed changes in vegetation states, and to take them into account in management policies

(Svenning and Sandel, 2013). Lag effects are also central to understanding resilience of an ecosystem (Holling, 1973; Walker et al., 2004).

Second, our findings imply a high priority and potential for managing fire-dependent ecosystems such as savannas. In these ecosystems, it has been argued that elevated $CO_2$ is the main driver for shrub encroachment and transitions to forest (Higgins and Scheiter, 2012; Midgley and Bond, 2015). Suitable management intervention can oppose $CO_2$ fertilization effects and

delay undesired vegetation changes (Scheiter and Savadogo, 2016). For instance, the introduction of fire can increase lags between transient and equilibrium vegetation states, whereas fire suppression for example by grazing (Pfeiffer et al., 2019) or fire management (Scheiter et al., 2015) can reduce disturbance-related lags. Other disturbances or land use activities such as herbivory or fuelwood harvesting have similar effects (Scheiter and Savadogo, 2016). Hence the potential for these ecosystems to persist in a disequilibrium state relative to climate and $CO_2$ creates the opportunity to mitigate changes brought about

by global change through management interventions. Conversely, allowing these systems to reach their equilibrium state has the potential to increase the global land carbon sink. While such management is relevant for carbon sequestration (Bastin





et al., 2019), it might, however, imply loss of biodiversity concomitant with losses of open savanna and grassland ecosystems (Veldman et al., 2015; Bond et al., 2019). Given the increasing lag size between transient and equilibrium vegetation states, management should decide at the local scale if current or desired vegetation states should be maintained as long as possible
or if ecosystems should be managed to account for vegetation changes expected in near future. Ignoring committed changes might imply rapid vegetation shifts that inhibit sustainable management actions.

Third, we found that the rate at which environmental conditions change determines the size of the lag between transient and equilibrium vegetation. Following the RCP 8.5 trajectory instead of the RCP 2.6 trajectory will therefore increase carbon debt both due to higher $CO_2$ mixing ratio and the acceleration of $CO_2$ enrichment in the atmosphere. Following the RCP 2.6
trajectory instead of the RCP 8.5 trajectory would decrease the maximum aboveground carbon debt in Africa from -60 PgC to -18 PgC in the absence of fire and from -47.5 PgC to -9 PgC in the presence of fire.

Finally, climate and greenhouse gas concentrations in the atmosphere are likely to change at an unprecedented rate (Prentice et al., 1993; Foster et al., 2017). Our results indicate that vegetation is most sensitive to changes in atmospheric $CO_2$ at the currently prevailing levels and values expected in near future (between approximately 350 ppm and 500 ppm, depending on
the simulation scenario and response variable investigated). Hence, we are currently in a period where small changes in $CO_2$ are likely to have large impacts on long-term vegetation change. The results also show that restoring savannas from heavily encroached wood-dominated states is a long process, particularly if fire is lost from these ecosystems. This finding raises the urgent need for society to act and reduce greenhouse gas emissions as the window of opportunity where human intervention can contribute to reverse climate change impacts might close soon.

## 4.3 Implications for vegetation modeling

The lag effects identified in our study have important implications for vegetation modelling and the process of testing and benchmarking models. Lag effects are prevalent both in results from transient model simulations using time series of climate data, and in data used for benchmarking, including remote sensing products (Saatchi et al., 2011; Simard et al., 2011; Avitabile et al., 2016) or data collected in field surveys. Previous studies identified sources of uncertainty in data-model comparisons
(Scheiter and Higgins, 2009; Langan et al., 2017) related to model uncertainties or data uncertainties. We argue that the presence of forcing and disturbance lags can add to disagreement between benchmarks and simulation results, such as modeled and satellite-derived productivity (Smith et al., 2016), carbon stocks, vegetation type, or species composition. Although DGVMs typically simulate transient vegetation states based on time series obtained from climate models, we argue that to improve the benchmarking process, we need to ensure that data and models represent similar successional stages. This can be achieved, e.g.
by applying appropriate model initialization methods using historical climate data or by considering effects of historic legacies on vegetation (Moncrieff et al., 2014). We concede that this is not an easy task as large-scale data on equilibrium vegetation states or lag sizes are typically not available, and as vegetation states have been modified by human land use for millennia. Remote sensing products such as the GEDI mission (gedi.umd.edu) may provide high-resolution data required to initialize models with biomass and vegetation structure.





The emergence of lag effects also highlights the relevance of adequate representation of demography, succession and disturbance regimes in vegetation models and makes a case for individual-based approaches. Understanding and quantifying lags necessitates prioritization and further model development with respect to these processes (Fisher et al., 2018) as well as improved knowledge of the rates at which these processes operate. Accurate representation of rates of changes may contribute to improve data-model agreement (Smith et al., 2016). In-depth model testing against available long-term observations of

successional changes in response to climate change and disturbance can further reduce model uncertainties. Fire or herbivore manipulation experiments conducted in Kruger National Park (Higgins et al., 2007) or in Burkina Faso (Savadogo et al., 2009) exemplify such data sets, in particular because they not only provide time series for benchmarking but also include relevant and documented disturbance processes under controlled conditions.

        In this study, we only considered natural fire regimes as simulated by aDGVM. However, to understand the full complexity

of vegetation lags, further disturbances need to be taken into consideration. This includes managed fire (intentionally planned as well as accidental anthropogenic fires), but also herbivory (grazing and browsing), fuelwood harvesting, deforestation or conversion of natural land to agricultural or forestry areas. In this study we only quantified lags in biome shifts, tree cover and aboveground carbon storage, but our framework can be extended to further ecosystem services in follow-up studies. In addition, we only considered changes in $CO_2$ in this study whereas changes in other key variables influencing vegetation,

particularly precipitation and temperature, were ignored. Previous aDGVM studies show that change in $CO_2$ is the main driver of simulated future vegetation change due to strong $CO_2$ fertilization effects (Scheiter and Higgins, 2009; Scheiter et al., 2015, 2018). We therefore expect that using time series of both $CO_2$ and climatic drivers following RCP scenarios will not change the fundamental results of our study. This expectation is supported by preliminary simulation results using various climatic drivers for RCP 8.5 and 4.5 (Pfeiffer *et al.*, unpublished).

**4.4   Further lags in the climate-vegetation system**

        In this study, we identified three lags between transient and equilibrium vegetation, namely forcing, disturbance and successional lags. Yet, due to the design of aDGVM our study ignores other lag effects in the climate-vegetation system.

        Migration lags occur due to a limited speed of seed dispersal and migration under variable climate. Dispersal and migration can, particularly in vegetation, typically not keep pace with climate change (Loarie et al., 2009). Migration lags have been

investigated, but typically using statistical approaches such as species distribution modelling (Thuiller et al., 2005). Extending DGVMs with dispersal models is, however, possible. For instance, Blanco et al. (2014) used a spatially explicit version of aDGVM to show that forest expansion rates in *Araucaria* forest-grassland mosaics in southern Brazil are sensitive to characteristics of the dispersal traits of *Araucaria* trees. Sato and Ise (2012) showed with SEIB-DGVM that including dispersal in projections of African vegetation until 2100 implies lagged responses in simulated biome boundary shifts.

Once a species has migrated into another suitable region, establishment lags due to competition between established and invading species can occur and prevent establishment of invading species. To quantify establishment lags in DGVMs, an accurate description of competition, demography and succession is necessary. Scheiter et al. (2013) argued that competition is often not adequately described in DGVMs. More advanced models are required that simulate competitive interactions between





individual plants for space, light, nutrients and water, and consider allelopathic interactions. In addition, novel approaches
based on trait variation such as aDGVM2 (Scheiter et al., 2013; Langan et al., 2017), JEDI-DGVM (Pavlick et al., 2013) or
LPJmL-FIT (Sakschewski et al., 2015) can be applied to identify establishment lags based on their detailed description of
individual plants and plant communities that are characterized by variable and dynamic traits instead of relying on a fixed
number of static plant functional types.

At longer time scales, evolutionary lags occur due to evolution of species adapted to changing environmental conditions or to
disturbance regimes (e.g., Simon et al., 2009; Guerrero et al., 2013). More advanced vegetation models are required to simulate
how evolutionary processes such as trait inheritance, mutation and cross-over allow plants to adapt to changing environmental
conditions over many generations. While aDGVM2 (Scheiter et al., 2013) includes these mechanisms, the model has so far not
been applied in an evolutionary context. Such an application would require a re-parametrization of mutation and cross-over
rates using empirical data to account for the temporal component of evolutionary processes.

Finally, atmosphere-biosphere lags can occur where delayed responses of vegetation change to environmental change feed
back to the environment to delay responses. For example, increasing forest cover due to $CO_2$ fertilization and resulting in-
crease in carbon sequestration may reduce $CO_2$ enrichment in the atmosphere and associated amplification of radiative forcing.
Investigation of atmosphere-biosphere lags requires fully coupled models that simulate biogeochemical fluxes between atmo-
sphere and biosphere. Jones et al. (2009) used such a fully coupled model to investigate lags in Amazon forest die-back, but
feed-backs to the climate system were not explicitly considered. Port et al. (2012) used the fully coupled MPI ESM to show
that lagged responses of vegetation may, in a scenario where $CO_2$ emissions are zero after 2120, reduce atmospheric $CO_2$ by
approximately 40ppm until 2300. Another well-studied example is the Sahel greening phenomenon where smooth changes in
rainfall regimes trigger abrupt and delayed responses in vegetation cover due to vegetation-atmosphere feed-backs (Brovkin
et al., 1998; Claussen et al., 1999; Foley et al., 2003).

**5 Conclusions**

To conclude, our study shows that vegetation generally lags behind changing atmospheric $CO_2$ mixing ratio. We are currently
in a phase of high carbon storage and tree cover debt and vegetation cover deviates substantially from the committed vegetation
state. Our study shows that vegetation is most sensitive to changes in atmospheric $CO_2$ at current levels of atmospheric $CO_2$
and those expected in near future. This finding indicates the need to act and reduce greenhouse gas emissions. Lags are larger
in fire-dependent systems such as savannas than in arid grasslands or forests. Lag effects in vegetation status need to be
considered for the development of management plans or mitigation strategies because we expect further changes in vegetation
even if emissions of $CO_2$ and other greenhouse gasses are reduced and the climate system stabilizes. There is an urgent need
to understand lag effects not only in response to variable $CO_2$, but also to other key variables of the climate system such as
temperature and precipitation, as well as to extreme events such as heat waves or drought.



*Code availability.* The aDGVM code as well as scripts to conduct the model experiments and analyze the results are available upon request. Please contact any of the authors.

*Author contributions.* SS, SH and GM conceived the study, SS conducted simulations and analyzed results, SS and MP created the figures, SS lead the writing with contributions of all co-authors.

*Competing interests.* We declare that no competing interests are present.

*Acknowledgements.* SS thanks the Deutsche Forschungsgemeinschaft (DFG) for funding (Emmy Noether grant SCHE 1719/2-1). MP thanks the German Federal Ministry of Education and Research (BMBF) for funding (SPACES initiative, 'SALLnet' project, grant 01LL1802B).





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

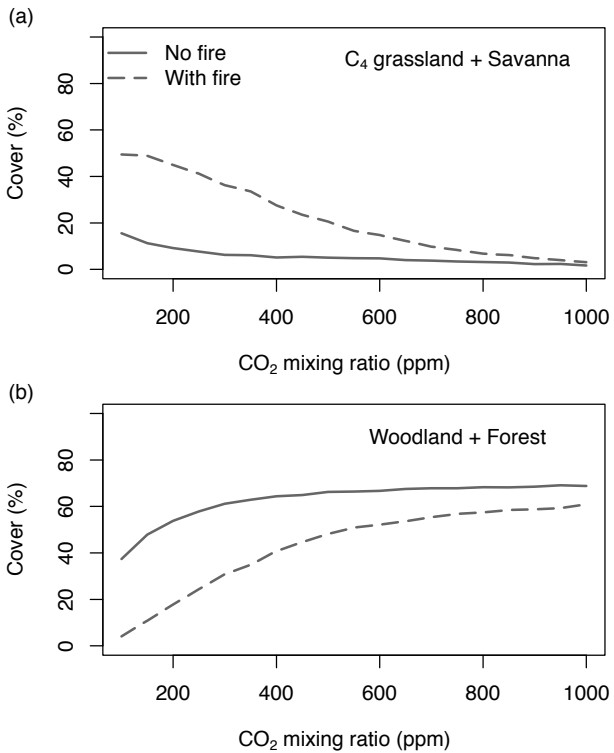

**Figure 1.** Area of Africa covered by (a) $C_4$-dominated ($C_4$ grassland and savanna) and (b) $C_3$-dominated (woodland and forest) vegetation under equilibrium conditions. Simulations were conducted until vegetation reached an equilibrium state. Differences between simulations without fire (solid lines) and with fire (dashed lines) indicate that both $C_3$ and $C_4$ vegetation states are possible. Fig. A1 shows cover fractions separated by biome type.



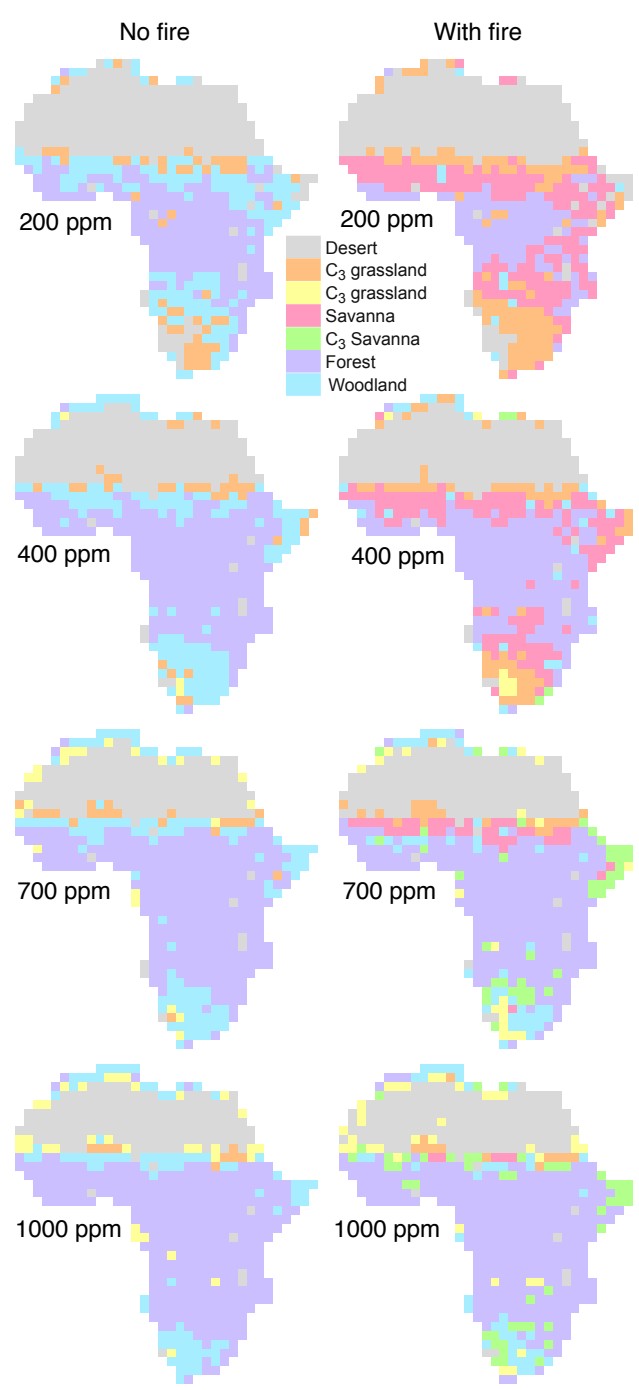

**Figure 2.** Biome distribution at different $CO_2$ mixing ratios and in the presence or absence of fire. Simulations were conducted until vegetation reached an equilibrium state.





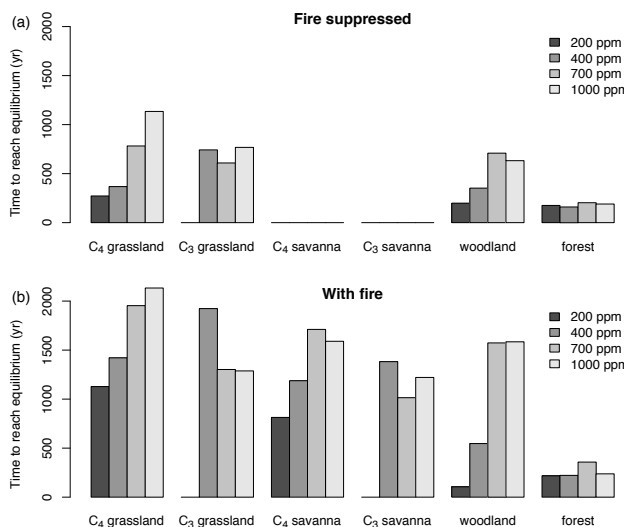

**Figure 3.** Time required to reach the equilibrium biome state in simulations with fixed $CO_2$ mixing ratio. Time was averaged for different biome types and different $CO_2$ mixing ratios. Times to reach equilibrium are shortest in forest, and not respond strongly to $CO_2$. In more open systems (grassland, savanna, woodland) times to reach equilibrium are longer than in forests, and equilibration times increase as $CO_2$ increases. Times to reach equilibrium are shorter under fire suppression (b) than in the presence of fire (a).





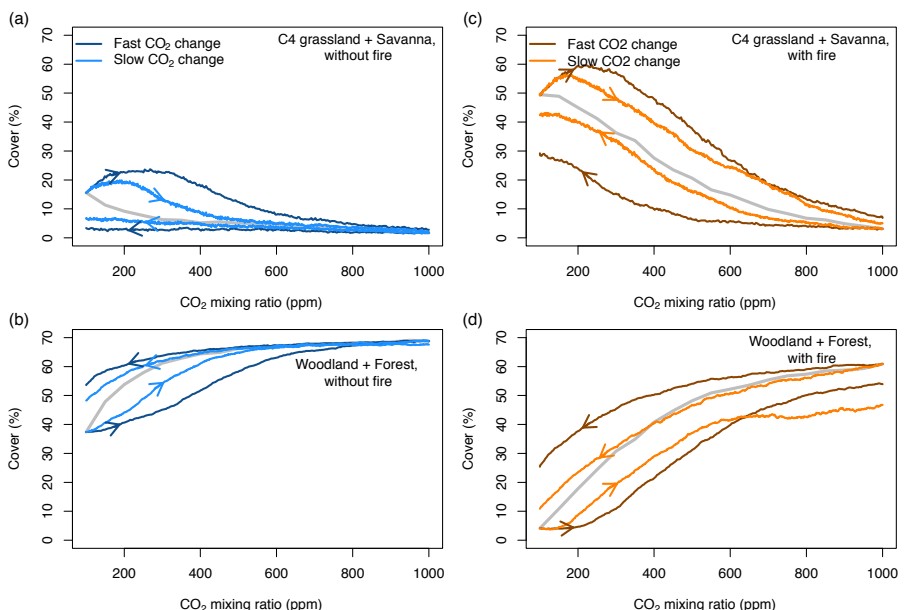

**Figure 4.** Percentages covered by $C_3$- and $C_4$-dominated vegetation in the presence and absence of fire. $CO_2$ is increased or decreased at two different rates between 100 ppm and 1000 ppm. The gray line indicates vegetation cover in equilibrium simulations. Arrows indicate whether $CO_2$ increases or decreases.


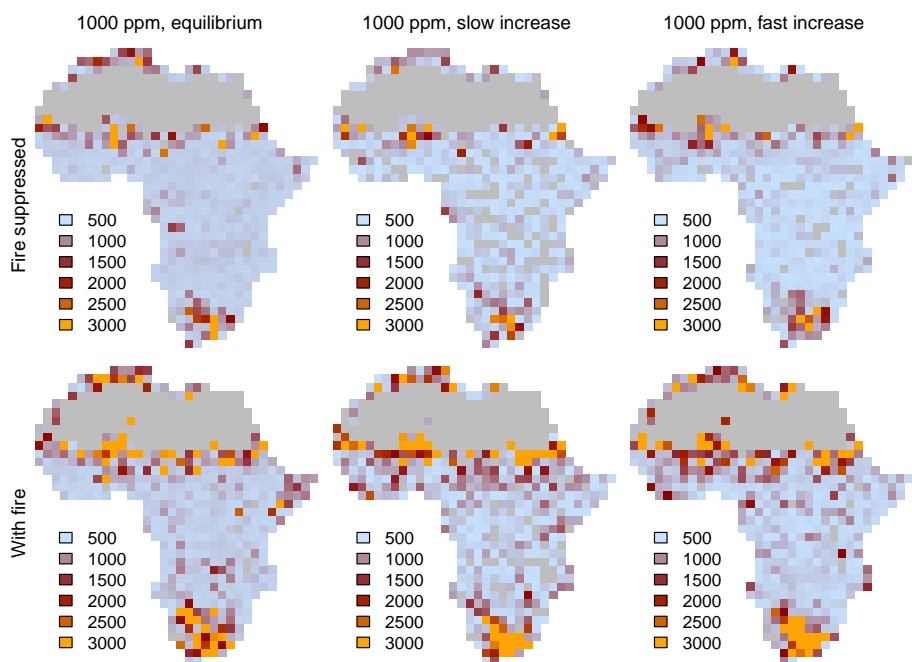

**Figure 5.** Time required to reach equilibrium in equilibrium simulations and transient simulations with both slow and fast increases of $CO_2$. Note that 3000 years in the legend means ≥3000 years, because simulations were run for a maximum of 3000 years.





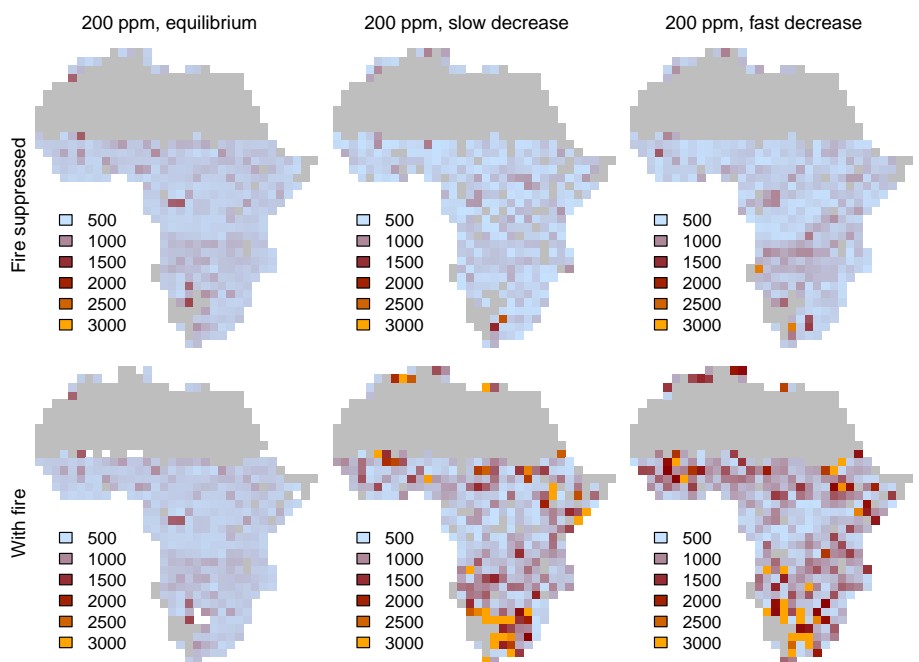

**Figure 6.** Time required to reach equilibrium in equilibrium simulations and transient simulations with both slow and fast decreases of $CO_2$. Note that 3000 years in the legend means ≥3000 years, because simulations were run for a maximum of 3000 years.





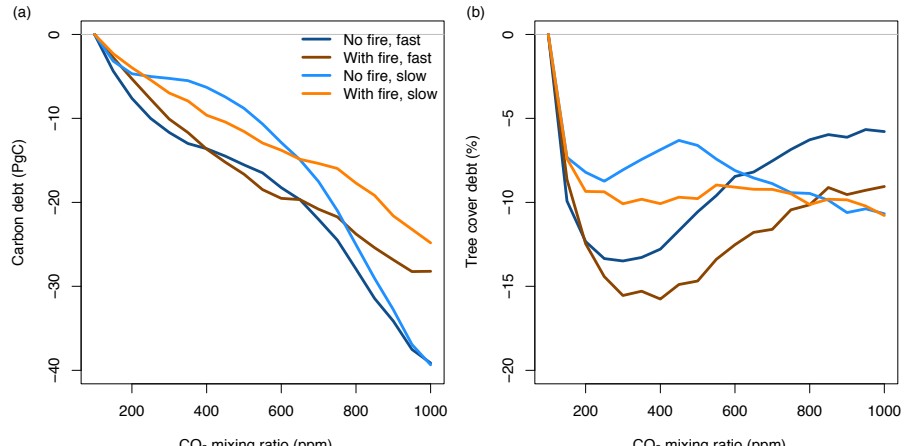

**Figure 7.** Debt of vegetation carbon and tree cover when the atmospheric $CO_2$ mixing ratio increases. Lines represent the mean of differences between transient and equilibrium simulations of all study sites in Africa (simulated at 2° resolution). See Fig. A2 for decreasing $CO_2$ and associated tree cover and carbon surplus.





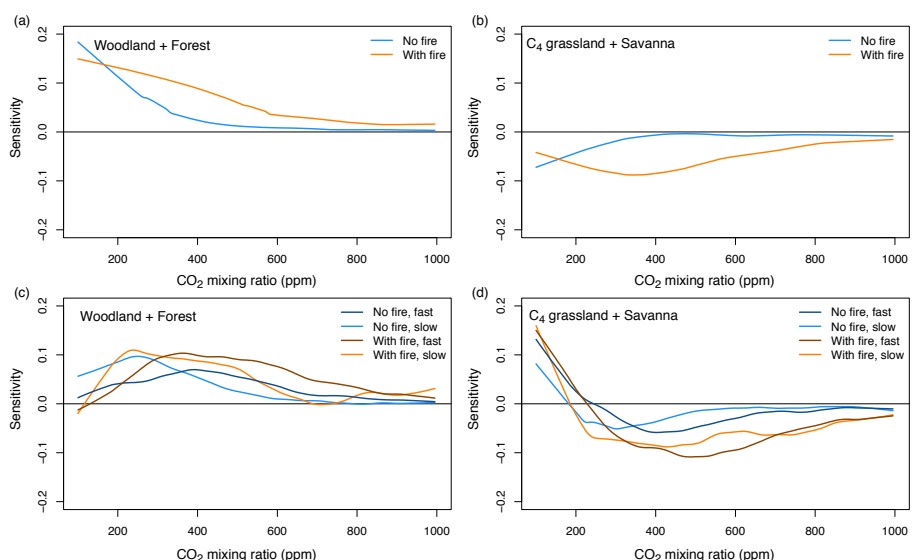

**Figure 8.** Sensitivity of vegetation cover to changes in the atmospheric $CO_2$ mixing ratio (in % change vegetation cover per ppm increase). Upper panels (a, b) show equilibrium simulations, lower panels (c, d) show transient simulations.



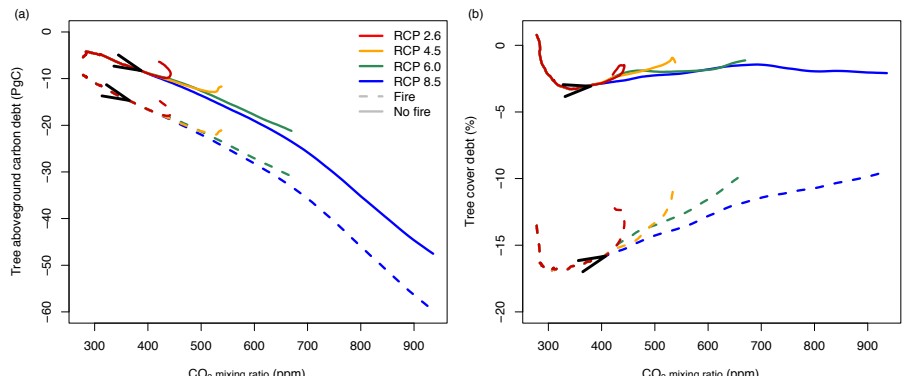

**Figure 9.** Vegetation carbon and tree cover debt when the atmospheric $CO_2$ mixing ratio increases according to different RCP scenarios. Lines represent differences between transient and equilibrium simulations averaged for all study sites in Africa (simulated at 2° resolution). Solid lines represent simulations without fire, dashed lines represent simulations with fire. Arrows indicate time between 1950 and 2100.

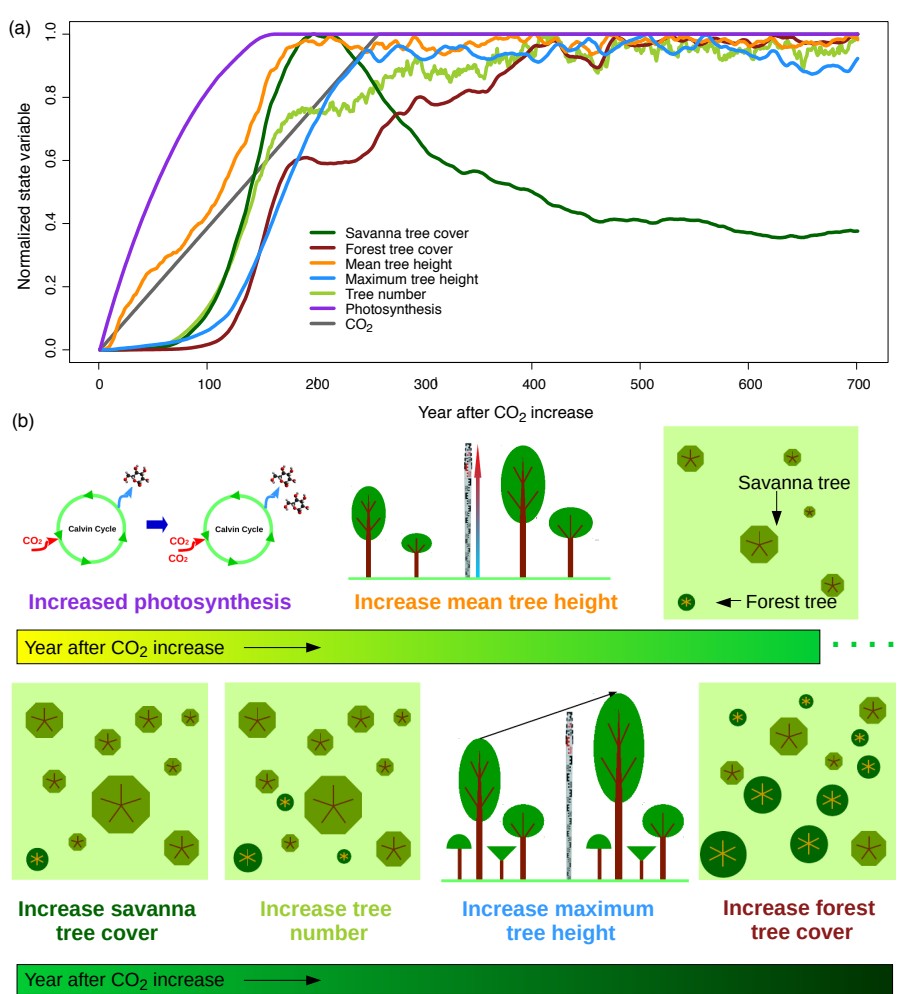

**Figure 10.** Vegetation responses to increasing $CO_2$. Panels show (a) time series of different state variables at a savanna study site in South Africa (26°S, 28°E), and (b) a schematic illustration of processes. State variables represent averages of 200 replicate simulation runs for the site. Normalization of state variables between zero and one based on minimum and maximum values was applied to be able to illustrate temporal lags between variables. Fig. A3 provides the time series without normalization and with units or respective variables.



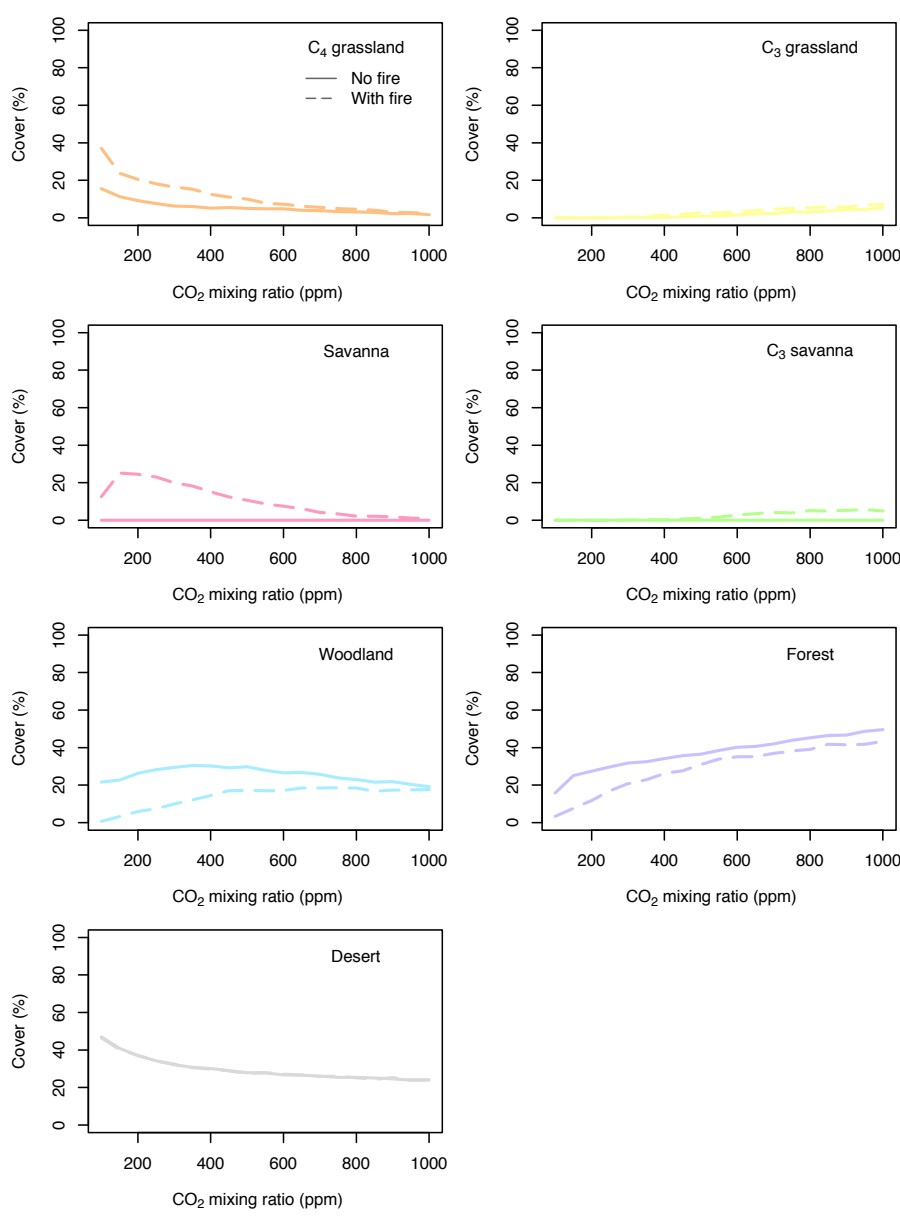

**Figure A1.** Area of Africa covered by different biome types. Simulations were conducted until vegetation reached an equilibrium state. Differences between simulations without fire (solid lines) and with fire (dashed lines) indicate that both $C_3$ and $C_4$ vegetation states are possible.





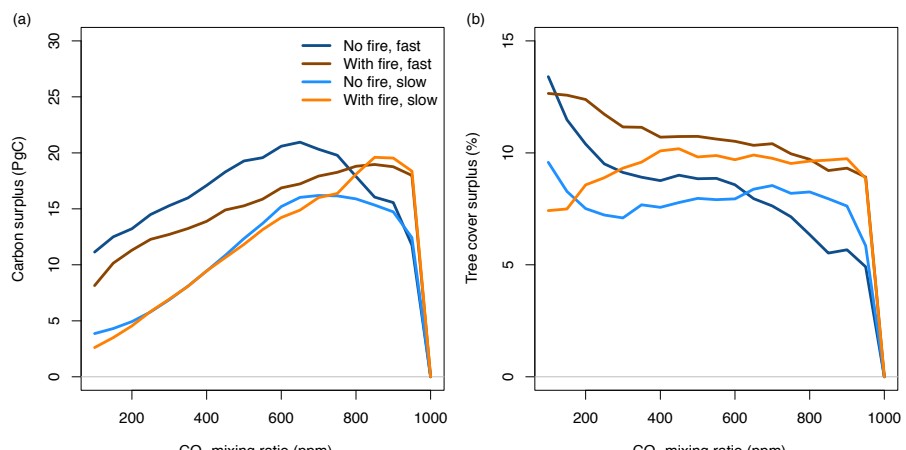

**Figure A2.** Surplus of tree cover and carbon when the atmospheric $CO_2$ mixing ratio decreases. Lines represent the mean of differences between transient and equilibrium simulations of all study sites in Africa (simulated at 2° resolution).



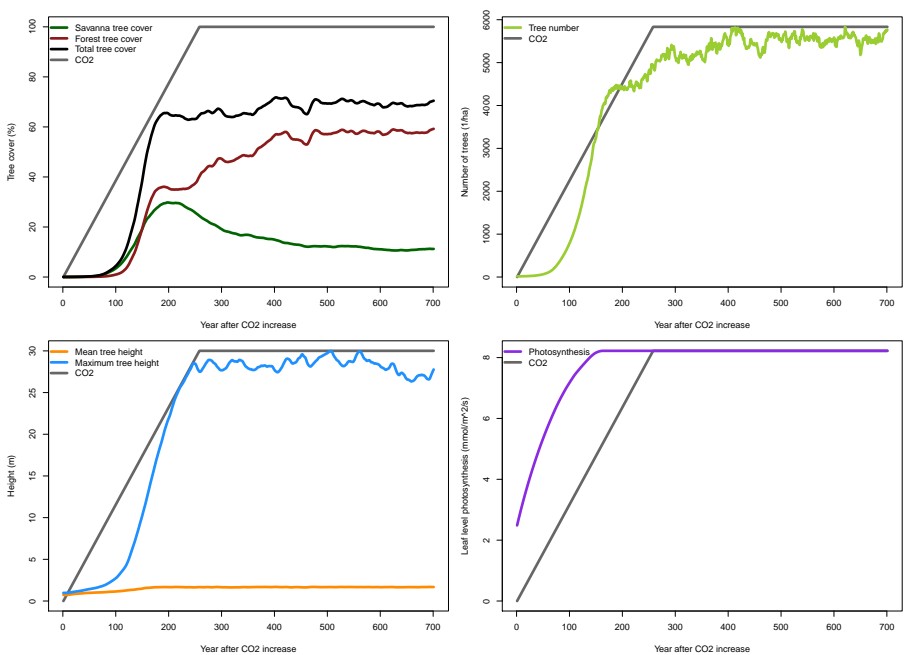

**Figure A3.** Time series of different state variables at a savanna study site in South Africa (26°S, 28°E). State variables represent averages of 200 replicate simulation runs for the site.