# Peer review of "African biomes are most sensitive to changes in CO2 under recent and near-future CO2 conditions"

_Biogeosciences, 2019_

## Referee Comment (RC1) · Anonymous Referee #1 · 15 Nov 2019

Schieter et al. provide a detailed assessment of the predictions of the aDGVM model in Africa focus on the question of timelags between transient and equilibrium climate states. They clearly illustrate the dependance of time lags and so-called carbon and tree cover 'debt' to the rates of change, the absolute values of CO2 and the presence or absence of fire. They link these findings back to applications in terms of assessing global carbon budgets, and implications for model benchmarking.

In general the paper is well written and the experimental design and results are both comprehensive and clearly depicted. I have few comments on these except for mostly minor grammatical points. On the discussion points I think there is a little room for

improvement. The notion of carbon debt needs a little more by way of explanation, specifically of why it is a debt and to whom the debt is owed, etc. The specific recommendation that the authors make to the land surface modeling community regardin forcing data, initialization strategies etc. are also a little vaguely worded and could do with some expansion.

Aside from that I think this manuscript provides a very solid illustration of the likely impact of lags on carbon cycle dynamics, and should provide a firm justification for further research into this topic.

Specific Comments

Line 12: Lag effects imply. (as opposed to implicate) Line 18: During Earth's history... Line 25: Needs a reference for RCP8.5. Line 49: Is 'mismatch' the right word? (given there's no real reason they should be matched in the first place. What about 'The substantial differences between the rate of change in environmental forcing and ..."? Line 52: Delete 'in' before 'constant' Line 63: Not sure I'd use 'oscillate' either. I'm not sure what I would use though, because isn't all vegetation to some extent in a transient successional state? Line 65: Influences 'whether', as opposed to 'if'. Line 90: I don't really understand this sentence... Line 100: I'm not sure that I totall ybuy that temperature increases are too heterogeneous to be interesting here? Maybe say that CO2 can be used as an illustration of the general principle which is likely extensible to shifts in other drivers? Line 103: What does 'typically' mean in this context? Line 112: Splitting hairs, but isn't it a contradiction to have a 'G' in aDGVM and then also say that it's outright only for savannas? 'Optimise for use in savannas' maybe? Line 121: In this era of 'trait-based' modeling, how is the shade tolerance implemented exactly? As an environmental limit on recruitment? Line 126: Should that be kJ/m2/s? Line 173: I don't quite understand here why you used all these different CO2 scenarios, or why you'd expect the vegetation to -not- be at equilibrium with constant forcing? Is it about cycling or oscillations in the intrinsic dynamics of the model? L178: Why was a smoothing algorithm needed? To smooth over stochastic outputs? L218: Were

there 200 replicates for all runs, or just for the study site in SA? (and if so, why?) L250: I really like this figure....Can you make it so that the text doesn't overlap the orange lines? L275: Why does fire slow down the transition? Is it because systems are stuck in stable equilibria one way or another? L285: The terminology of 'debt' is slightly confusing to me, especially as the quantities of 'debt' are negative. Does that mean there is negative debt? And to whom is the debt owed? The atmosphere or the ecosystem? I guess a negative carbon debt is a promise that the system will take up more CO2 in the future? L315: I'm tripped up again by the use of 'typically'... Do you mean in the transient simulations? By implication in the real world? Also technically I guess this is a prediction, rather than an illustration, so many change 'show' to something less forceful. L320: Also change 'previous findings' to 'previous model results' or suchlike. L325: I feel like this section is missing a statement on what, if any, other studies have had to say on this topic, for Africa or elsewhere. Or indeed, is this type of analysis totally novel? L328: This justification for why the fire runs have greater lags should perhaps come earlier in the manuscript. L358: This needs a little more clarification. I.e. "(the strong CO2 effect) can compensate for other predicted changes in climate drivers, e.g. reduced rainfall" L357: Maybe expand on what a high CO2 sensitiviy might mean for the results of this paper? L385: change to :prevents both savanna and forest trees recruitment" L398: Need a reference for 'as is currently often the case'. L420: Would it make sense to frame the carbon debt here in terms of the overall carbon budgets of the different RCPs? For example, if emissions stabilises at the end of RCPxx, then the ecosystems would continue to keep absorbing carbon to get to their equilibrium ecosystem state and re-absorb xx PgC from the system? L439: I don't really understand this part. How can modifications to the initial climate conditions have an impact on the successional stage? Also, I think it's fairly standard to do pre-industrial spin-up and then transient simulations to the present day. Are you proposing an alternative approach to climate drivers here? L444: Further, given that we need to run transient simulations with ramping CO2, that is slightly at odds with initializing with contemporary observations. L474: Maybe cite Julia Nabels work with

TREEMIG as illustrative of the complexities of the implementation of seed dispersal? L500: Again, I'd argue that it 'predicts' or 'projects' or 'indicates', but perhaps not that it 'shows'.

---

## Referee Comment (RC3) · Anonymous Referee #2 · 27 Nov 2019

General Comments

This paper considers the lag between a transient and committed vegetation state under changing CO2 and due to the disturbance effect of fire, using the aDGVM. In my view the paper is well written, clearly structured, and presents relevant and interesting results. The study is structured around 4 hypotheses which consider the current vegetation state, the impact of rate of change, the extent of change, and the effects of fire, which are novel and useful. The methods are explained clearly. The definition of equilibrium presented in equation 1 appears logical, although I wonder if there is already a published method for this that has been used in other studies. The results

are presented in a logical way, and the text supports the figures throughout. I believe the conclusions are a valid interpretation of the results and that they are substantial and useful. I have some small comments on specific sections as outlined below, but otherwise I think the paper is of very good quality.

Specific Comments

Line 19 – Include the time period for the Devonian period to give context

Line 28 - Paleocene-Eocene Thermal Maximum (PETM), a period with high carbon emissions some 56 million years ago – It would be nice to see a little more about this period and explain why the carbon emissions were high

Line 50 – is there a reference for this definition of equilibrium? I wonder if there is another method available which has been used in already published studies that can be referred to. I can see the logic of this method but some extra reference to existing methodology, and why it has been altered if necessary, would make this stronger

Section 2.1 Line 110 – There aren't many PFTs represented in aDGVM. However it is mentioned in the discussion that this may cause an underestimation in lag time in forests, and as the study is focused on one savanna location I think it is enough for this study

Line 136 – the performance of aDGVM has been evaluated in terms of vegetation, but what about fire? It would be good to see some evidence that the fire model is reliable, at least for the location picked

Section 3.6 Line 300 – can you give an explanation as to why the carbon debt continues to increase when the tree cover debt decreases?

Fig 3 Bar plot – if fire is suppressed in forests (L384) would you not expect the forest results in figure 3 a and b to be the same, or would there still be some fire?

Also from figure 3, I think it would be worth quantifying the lag time and noting in the

abstract how much longer it takes to reach equilibrium per X increase in CO2, which is an important result

Line 256 – Lags are larger at low and intermediate CO2 mixing ratios and decrease at higher CO2. How does this fit with 'The time until vegetation reaches an equilibrium state. . .. Increase[s] with CO2' (L236)

Line 270 / Figure 5 and 6 – It follows that the time taken for the transient simulations to reach equilibrium is measured, but how is the time taken to reach equilibrium in equilibrium simulations measured? In other words what is the equilibrium simulation initialised from?

Line 289 – I think specifying that the debt is "larger" would be better than "higher" given the values are increasingly negative

Technical Comments

Line 18 – Earth's history?

Line 25 – Define RCP (Representative Concentration Pathway)

Line 91 – Does the a in aDGVM stand for anything?

Line 116 - "This approach allows to model how herbivores" – allows us to model?

Line 228 – "C4 or C3-dominated vegetation if fire is present or absent" respectively.

In most of the figures C4 grassland and savanna is labelled, but woodland and forest is not labelled as C3 despite being referred to in the text as C3

---

## Author Comment (AC1) · 18 Dec 2019

This review is identical to review RC2 but referee #1 (but unformatted). We therefore only respond to review RC2 by referee #1.

---

## Author Comment (AC2) · 18 Dec 2019

Schieter et al. provide a detailed assessment of the predictions of the aDGVM model in Africa focus on the question of timelags between transient and equilibrium climate states. They clearly illustrate the dependance of time lags and so-called carbon and tree cover 'debt' to the rates of change, the absolute values of CO2 and the presence or absence of fire. They link these findings back to applications in terms of assessing global carbon budgets, and implications for model benchmarking.

In general the paper is well written and the experimental design and results are both comprehensive and clearly depicted. I have few comments on these except for mostly

minor grammatical points. On the discussion points I think there is a little room for improvement. The notion of carbon debt needs a little more by way of explanation, specifically of why it is a debt and to whom the debt is owed, etc.

RESPONSE: We will clarify the notation of carbon debt, see our response below.

The specific recommendation that the authors make to the land surface modeling community regarding forcing data, initialization strategies etc. are also a little vaguely worded and could do with some expansion.

RESPONSE: We will clarify recommendations regarding model forcing and initialization. See also our response below.

Aside from that I think this manuscript provides a very solid illustration of the likely impact of lags on carbon cycle dynamics, and should provide a firm justification for further research into this topic.

RESPONSE: Thank you for the positive feedback.

Specific Comments Line 12: Lag effects imply. (as opposed to implicate)

RESPONSE: We will replace implicate by imply.

Line 18: During Earth's history. . .

RESPONSE: We will modify as suggested.

Line 25: Needs a reference for RCP8.5.

RESPONSE: We will add a reference (Meinshausen et al. 2011).

Line 49: Is 'mismatch' the right word? (given there's no real reason they should be matched in the first place. What about 'The substantial differences between the rate of change in environmental forcing and . . .'' ?

RESPONSE: We agree and we will modify the text as suggested.

Line 52: Delete 'in' before 'constant'

RESPONSE: We will delete 'in'.

Line 63: Not sure I'd use 'oscillate' either. I'm not sure what I would use though, because isn't all vegetation to some extent in a transient successional state?

RESPONSE: We will reword and state 'vegetation is regularly forced into early successional states'.

Line 65: Influences 'whether', as opposed to 'if'.

RESPONSE: We will reword as suggested and use 'whether'.

Line 90: I don't really understand this sentence. . .

RESPONSE: We will reword the sentence: 'Yet, previous studies often focused on CO2 levels predicted for 2100, assuming that both CO2 and the climate system will have stabilized by then. Here, we study lag effects for a CO2 gradient ranging from pre-industrial to future levels.'

Line 100: I'm not sure that I totally buy that temperature increases are too heterogeneous to be interesting here? Maybe say that CO2 can be used as an illustration of the general principle which is likely extensible to shifts in other drivers?

RESPONSE: We agree that consideration of temperature and precipitation is also important and interesting. Therefore, a study considering these variables is in preparation. A further reason why we ignored these variables is that for most of the simulations, we study a CO2 gradient from 100ppm to 1000pmm and it is challenging (or even impossible) to find time series for temperature and precipitation that cover and correspond to such a large range. Hence, studies considering several climatic variables and CO2 are constrained to a much smaller range (gridded products providing continuous time series are often constrained to the period 1950 to 2099). We will include this justification as well as the suggestion of the referee (CO2 can be used as illustration

of the general principles) in the revision. In the Discussion we already mention that simulations considering various climate variables are necessary.

Line 103: What does 'typically' mean in this context?

RESPONSE: We will reword this and state 'vegetation is, in all transient scenarios that we consider, not in equilibrium . . .'.

Line 112: Splitting hairs, but isn't it a contradiction to have a 'G' in aDGVM and then also say that it's outright only for savannas? 'Optimise for use in savannas' maybe?

RESPONSE: aDGVM was originally developed for savannas, but with the intention to extend it to the global scale. Therefore, the name contains the 'G'. As suggested we will reword to '. . . dynamic vegetation model optimized for tropical grass-tree systems'.

Line 121: In this era of 'trait-based' modeling, how is the shade tolerance implemented exactly? As an environmental limit on recruitment?

RESPONSE: We assume that forest trees are less affected by light competition than savanna trees, such that forest trees have higher growth rates in dense vegetation stands with intense light competition than savanna trees. In aDGVM we use parameters that describe the strength of light competition between different grass and tree types, and we used different parameters for savanna and forest trees. We will add a statement to clarify this: 'Shade tolerance is implemented by different effects of light availability, which is in turn influenced by competitor plants, on tree growth rates. Fire tolerance is implemented by different topkill functions and re-sprouting probabilities after fire.'

Line 126: Should that be kJ/m2/s?

RESPONSE: Fireline intensity usually given in kW per meter flaming front, which is equal to kJ/m/s. We therefore think that units in the manuscript are correct.

Line 173: I don't quite understand here why you used all these different CO2 scenarios,

or why you'd expect the vegetation to -not- be at equilibrium with constant forcing? Is it about cycling or oscillations in the intrinsic dynamics of the model?

RESPONSE: We expect that in transient runs, vegetation is not in equilibrium with the environment (i.e. $CO_2$) because exposure to these $CO_2$ levels is not long enough to allow vegetation to reach equilibrium. We therefore require all the equilibrium simulations for different $CO_2$ levels that are transgressed during transient simulations, to determine the deviance between vegetation states at transient vs equilibrium runs. In equilibrium runs, we do expect that vegetation is in equilibrium with constant forcing. This also includes equilibrium states where state variables oscillate regularly, e.g., due to fire impacts. The different $CO_2$ scenarios with constant $CO_2$ forcing are required to identify equilibrium vegetation states for a variety of constant environmental (i.e. $CO_2$) conditions. In addition, we simulated transient scenarios with increasing and decreasing $CO_2$ and different rates of change. We will carefully check and reword the text to make clear why we require all these simulations.

L178: Why was a smoothing algorithm needed? To smooth over stochastic outputs?

RESPONSE: Simulations for equilibrium conditions were only conducted for a set of $CO_2$ concentrations (100ppm, 150ppm, 200ppm, . . ., 1000ppm). In contrast, transient simulations provide model results for a much higher resolution of $CO_2$ levels between 100ppm and 1000ppm (depending on the rate of change of $CO_2$). To be able to calculate differences between transient and equilibrium vegetation states for the entire $CO_2$ gradient, we not only need values at 100, 150, 200ppm, . . . in equilibrium simulations, but also values in between. Therefore, we applied the smoothing algorithm. In addition, as mentioned by the referee, smoothing also reduces stochasticity in model outputs. We will reword the text to clarify why we apply smoothing.

L218: Were there 200 replicates for all runs, or just for the study site in SA? (and if so, why?)

RESPONSE: The 200 replicates were only done for the study site, but not for

continental-scale simulations. We conducted replicate runs because processes such as fire occurrence or demographic processes are stochastic in aDGVM. Averaging across these replicates allows us to reduce stochasticity and to obtain more robust responses of various state variables. For continental-scale simulations, we typically consider responses for different biomes, i.e., we average over space. Replicate simulation runs were therefore not conducted. We will reword the methods to make clear that and why we only conducted one run for continental-scale simulations.

L250: I really like this figure. . ..Can you make it so that the text doesn't overlap the orange lines?

RESPONSE: We will modify Fig 4 to avoid overlap between text and lines, and we will also check other figures to avoid overlap.

L275: Why does fire slow down the transition? Is it because systems are stuck in stable equilibria one way or another?

RESPONSE: Yes, we argue in the discussion that fire prevents vegetation transitions and keeps vegetation in the fire-driven state. We will also include a short explanation in the results section: 'Increased lag size in the presence of fire can be attributed to hysteresis effects. High fire activity traps vegetation in a fire-driven state and prevents biome transitions into alternative vegetation states.'

L285: The terminology of 'debt' is slightly confusing to me, especially as the quantities of 'debt' are negative. Does that mean there is negative debt? And to whom is the debt owed? The atmosphere or the ecosystem? I guess a negative carbon debt is a promise that the system will take up more $CO_2$ in the future?

RESPONSE: Debt refers to carbon storage potential of vegetation that has not been realized yet. It indicates that vegetation has the potential to sequester more carbon under given $CO_2$ levels than transient simulation suggest. The potential carbon storage for given $CO_2$ is defined by the equilibrium simulations. In figures, we plotted debt with

negative values to clearly distinguish from surplus (Fig A2) and to make clear that vegetation is committed to substantial changes even if the climate system stabilizes. We will carefully revise the definition of carbon debt in the revision (in sec. 3.4) to make clear that 'we define debt is carbon storage potential that has not been realized yet, and that the atmosphere owes to vegetation'. We prefer to keep the negative sign to make clear that potential carbon and tree cover exceed carbon and tree cover in transient runs.

L315: I'm tripped up again by the use of 'typically'... Do you mean in the transient simulations? By implication in the real world? Also technically I guess this is a prediction, rather than an illustration, so many change 'show' to something less forceful.

RESPONSE: Yes, we mean that vegetation in transient simulations is not in equilibrium. We will reword to avoid 'typically' and state 'Using a dynamic vegetation model, we predict that vegetation exposed to transient environmental forcing is not in equilibrium with environmental conditions . . .'. In the discussion, we also provide references reporting lag effects in empirical experiments.

L320: Also change 'previous findings' to 'previous model results' or suchlike.

RESPONSE: We will reword as suggested.

L325: I feel like this section is missing a statement on what, if any, other studies have had to say on this topic, for Africa or elsewhere. Or indeed, is this type of analysis totally novel?

RESPONSE: The aim of this paragraph is to put the results presented in this study into the context of previous aDGVM studies and highlight the novelty within the 'aDGVM world'. Yet, we discuss the novel results in the context of previous studies in later sections of the discussion. We therefore prefer to keep this section focused on aDGVM. Yet we will carefully reword the section to make clear that it only refers to aDGVM results.

L328: This justification for why the fire runs have greater lags should perhaps come earlier in the manuscript.

RESPONSE: We will add an explanation why fire increases lags in the results. See also our response to comments regarding L275.

L358: This needs a little more clarification. I.e. "(the strong CO2 effect) can compensate for other predicted changes in climate drivers, e.g. reduced rainfall"

RESPONSE: We will reword as suggested.

L357: Maybe expand on what a high CO2 sensitiviy might mean for the results of this paper?

RESPONSE: We will check the manuscript carefully to ensure that implications are highlighted sufficiently. In particular, we will add 'If aDGVM overestimates CO2 sensitivity of vegetation, the size of carbon debt due to lag effects may be overestimated, while the duration of lags may be underestimated if simulated response to changing CO2 is more sensitive than in reality. We are however confident that even with reduced CO2 sensitivity the overall response pattern would remain rubust, although the quantities might change.'

L385: change to :prevents both savanna and forest trees recruitment"

RESPONSE: We will reword this statement (in response to referee 3): "Fire rarely occurs in simulated forests, and therefore they reach equilibrium faster than other biome types. Fire activity in forests is sufficient to slightly increase times to reach equilibrium in comparison to simulations with fire suppressed."

L398: Need a reference for 'as is currently often the case'.

RESPONSE: We will delete 'as is currently often the case', because this statement is very bold.

L420: Would it make sense to frame the carbon debt here in terms of the overall

carbon budgets of the different RCPs? For example, if emissions stabilises at the end of RCPxx, then the ecosystems would continue to keep absorbing carbon to get to their equilibrium ecosystem state and re-absorb xx PgC from the system?

RESPONSE: Good suggestion, we will reword and reframe the paragraph as suggested.

L439: I don't really understand this part. How can modifications to the initial climate conditions have an impact on the successional stage? Also, I think it's fairly standard to do pre-industrial spin-up and then transient simulations to the present day. Are you proposing an alternative approach to climate drivers here?

RESPONSE: We argue that initial vegetation states (rather than climate) in the model should agree with successional states of vegetation at the beginning of the transient simulation run. This can for example be achieved by considering land-use history, or by ensuring that vegetation height or age is in agreement with observation and not only variables such biomass, NPP, GPP. An approach using forest height for initialization has for example been adopted by Rödig et al. (2017) for initialization of the FORMIND model. We will carefully revise this paragraph to be more specific about the initialization procedures.

Rödig, E, Cuntz, M, Heinke, J, Rammig, A, Huth, A. (2017) Spatial heterogeneity of biomass and forest structure of the Amazon rain forest: Linking remote sensing, forest modelling and field inventory. Global Ecol Biogeogr. 26: 1292– 1302.

L444: Further, given that we need to run transient simulations with ramping CO2, that is slightly at odds with initializing with contemporary observations.

RESPONSE: We don't see a contradiction here, because initialization can also consider transient simulations. For example one could use simulated and observed time-series of LAI or NDVI for model benchmarking. Such benchmarking would require the use of transient climate and CO2 forcing.

L474: Maybe cite Julia Nabels work with TREEMIG as illustrative of the complexities of the implementation of seed dispersal?

RESPONSE: We will cite the Nabel et al. (2013) Ecological Modelling paper.

L500: Again, I'd argue that it 'predicts' or 'projects' or 'indicates', but perhaps not that it 'shows'.

RESPONSE: As suggested, we will replace 'shows' in the Conclusions.

———————————————————

---

## Author Response (AR1)

Dear Editor,

please find the revised version of the manuscript. We carefully considered all the comments and suggestions by the referee, revised the manuscript accordingly, and provide point-by-point responses to all comments. Our revisions are highlighted in bold font in the manuscript. In addition, we checked the for typos and reworded the text where necessary.

We are looking forward to your decision,

Best regard,
Simon Scheiter and co-authors

**Responses to RC1 by referee #1**

This review is identical to review RC2 but referee #1. We therefore only respond to review RC2.

**Responses to RC2 by referee #1**

Schieter et al. provide a detailed assessment of the predictions of the aDGVM model in Africa focus on the question of timelags between transient and equilibrium climate states. They clearly illustrate the dependance of time lags and so-called carbon and tree cover 'debt' to the rates of change, the absolute values of CO2 and the presence or absence of fire. They link these findings back to applications in terms of assessing global carbon budgets, and implications for model benchmarking.

In general the paper is well written and the experimental design and results are both comprehensive and clearly depicted. I have few comments on these except for mostly minor grammatical points. On the discussion points I think there is a little room for improvement. The notion of carbon debt needs a little more by way of explanation, specifically of why it is a debt and to whom the debt is owed, etc.

**RESPONSE**: We clarified the notation of carbon debt, see our response below.

The specific recommendation that the authors make to the land surface modeling community regarding forcing data, initialization strategies etc. are also a little vaguely worded and could do with some expansion.

**RESPONSE**: We clarified recommendations regarding model forcing and initialization. See also our response below.

Aside from that I think this manuscript provides a very solid illustration of the likely impact of lags on carbon cycle dynamics, and should provide a firm justification for further research into this topic.

**RESPONSE**: Thank you for the positive feedback.

Specific Comments
Line 12: Lag effects imply. (as opposed to implicate)

**RESPONSE**: We replaced 'implicate' by 'imply'.

Line 18: During Earth's history. . .

**RESPONSE**: We modified as suggested and added the 's'.

Line 25: Needs a reference for RCP8.5.

**RESPONSE**: We added a reference (Meinshausen et al. 2011).

Line 49: Is 'mismatch' the right word? (given there's no real reason they should be matched in the first place. What about 'The substantial differences between the rate of change in environmental forcing and . . ." ?

**RESPONSE**: We agree and we modified the text as suggested.

Line 52: Delete 'in' before 'constant'

**RESPONSE**: We deleted 'in'.

Line 63: Not sure I'd use 'oscillate' either. I'm not sure what I would use though, because isn't all vegetation to some extent in a transient successional state?

**RESPONSE**: We reworded the sentence and now state 'Abrupt and repeated disturbances imply that vegetation is regularly forced into early or intermediate successional states'.

Line 65: Influences 'whether', as opposed to 'if'.

**RESPONSE**: We reworded as suggested and use 'whether'.

Line 90: I don't really understand this sentence. . .

**RESPONSE**: We reworded the sentence: 'Yet, previous studies often focused on $CO_2$ levels predicted for 2100, assuming that both $CO_2$ and the climate system will have stabilized by then. Studies on lag effects for a $CO_2$ gradient ranging from pre-industrial to future levels are, however, rare.'

Line 100: I'm not sure that I totally buy that temperature increases are too heterogeneous to be interesting here? Maybe say that $CO_2$ can be used as an illustration of the general principle which is likely extensible to shifts in other drivers?

**RESPONSE**: We agree that consideration of temperature and precipitation is also important and interesting. Therefore, a study considering these variables is in preparation. A further reason why we ignored these variables is that for most of the simulations, we study a $CO_2$ gradient from 100ppm to 1000pmm and it is challenging (or even impossible) to find time series for temperature and precipitation that cover and correspond to such a large range. Hence, studies considering several climatic variables and $CO_2$ are constrained to a much smaller range (gridded products providing continuous time series are often constrained to the period 1950 to 2099). We included this

justification as well as the suggestion of the referee: 'Datasets containing continuous time series of CO2 between pre-industrial and future levels and associated climate are rare. While precipitation, temperature, and other environmental variables influence ecosystems, in this study we focus on CO2 effects. We argue that CO2 is sufficient to illustrate the general principles underlying lags between environmental conditions and vegetation.'
In the Discussion we already mention that simulations considering various climate variables are necessary and in preparation.

Line 103: What does 'typically' mean in this context?

RESPONSE: We reworded this and state 'vegetation is, in all transient scenarios that we consider, not in equilibrium …'.

Line 112: Splitting hairs, but isn't it a contradiction to have a 'G' in aDGVM and then also say that it's outright only for savannas? 'Optimise for use in savannas' maybe?

RESPONSE: aDGVM was originally developed for savannas, but with the intention to extend it to the global scale. Therefore, the name contains the 'G'. We considered to replace 'developed' by 'optimized' as suggested. However, 'optimized' might suggest that we used numerical optimization methods to tune and calibrate the model to savanna ecosystems, but this was not done (at least for the model version used in this study). We therefore decided to keep 'developed for savannas'.

Line 121: In this era of 'trait-based' modeling, how is the shade tolerance implemented exactly? As an environmental limit on recruitment?

RESPONSE: We assume that forest trees are less affected by light competition than savanna trees, such that forest trees have higher growth rates in dense vegetation stands with intense light competition than savanna trees. In aDGVM we use parameters that describe the strength of light competition between different grass and tree types, and we used different parameters for savanna and forest trees. We added a statement to clarify this: 'Shade tolerance is implemented by different effects of light availability on tree growth rates. Light availability is in turn influenced by competitor plants. Fire tolerance is implemented by different topkill functions and re-sprouting probabilities after fire.'

Line 126: Should that be kJ/m2/s?

RESPONSE: Fireline intensity is usually given in kW per meter flaming front, which is equal to kJ/m/s. The units in the manuscript are therefore correct.

Line 173: I don't quite understand here why you used all these different CO2 scenarios, or why you'd expect the vegetation to -not- be at equilibrium with constant forcing? Is it about cycling or oscillations in the intrinsic dynamics of the model?

RESPONSE: We expect that in transient runs, vegetation is not in equilibrium with the environment (i.e. CO2) because exposure to these CO2 levels is not long enough to allow vegetation to reach equilibrium. We therefore require all the equilibrium simulations for different CO2 levels that are transgressed during transient simulations, to determine the deviance between vegetation states at transient vs equilibrium runs. In equilibrium runs, we do expect that vegetation is in equilibrium with

constant forcing. This also includes equilibrium states where state variables oscillate regularly, e.g., due to fire impacts. The different CO2 scenarios with constant CO2 forcing are required to identify equilibrium vegetation states for a variety of constant environmental (i.e. CO2) conditions. In addition, we simulated transient scenarios with increasing and decreasing CO2 and different rates of change. We checked and reworded the text to make clear why we require all these simulations.

L178: Why was a smoothing algorithm needed? To smooth over stochastic outputs?

**RESPONSE**: Simulations for equilibrium conditions were only conducted for a set of CO2 concentrations (100ppm, 150ppm, 200ppm, …, 1000ppm). In contrast, transient simulations provide model results for a much higher resolution of CO2 levels between 100ppm and 1000ppm (depending on the rate of change of CO2). To be able to calculate differences between transient and equilibrium vegetation states for the entire CO2 gradient, we not only need values at 100, 150, 200ppm, … in equilibrium simulations, but also values in between. Therefore, we applied the smoothing algorithm. In addition, as mentioned by the referee, smoothing also reduces stochasticity in model outputs. We reworded the text to clarify why we apply smoothing.

L218: Were there 200 replicates for all runs, or just for the study site in SA? (and if so, why?)

**RESPONSE**: The 200 replicates were only done for the study site, but not for continental-scale simulations. We conducted replicate runs because processes such as fire occurrence or demographic processes are stochastic in aDGVM. Averaging across these replicates allows us to reduce stochasticity and to obtain more robust responses of various state variables. For continental-scale simulations, we typically consider responses for different biomes, i.e., we average over space. Replicate simulation runs were therefore not conducted due to limitations on computation time. For the continental scale simulations we added: 'For continental-scale simulations, we only conducted one model run for each scenario, but no replicates. This single run is sufficient, as we aggregate model results per biome in our analyses.'

L250: I really like this figure. . ..Can you make it so that the text doesn't overlap the orange lines?

**RESPONSE**: We modified Fig 4 to avoid overlap between text and lines.

L275: Why does fire slow down the transition? Is it because systems are stuck in stable equilibria one way or another?

**RESPONSE**: Yes, we argue in the discussion that fire prevents vegetation transitions and keeps vegetation in the fire-driven state. We also included a short explanation in the results section: 'Longer times in the presence of fire can be attributed to hysteresis effects. High fire activity traps vegetation in a fire-driven state and prevents biome transitions into alternative vegetation states.'

L285: The terminology of 'debt' is slightly confusing to me, especially as the quantities of 'debt' are negative. Does that mean there is negative debt? And to whom is the debt owed? The atmosphere or the ecosystem? I guess a negative carbon debt is a promise that the system will take up more CO2 in the future?

**RESPONSE**: Debt refers to carbon storage potential of vegetation that has not been realized yet. It indicates that vegetation has the potential to sequester more carbon under given CO2 levels than

transient simulation suggest. The potential carbon storage for given $CO_2$ is defined by the equilibrium simulations. In figures, we plotted debt with negative values to clearly distinguish from surplus (Fig A2) and to make clear that vegetation is committed to substantial changes even if the climate system stabilizes. We revised the definition of carbon debt in the revision (in sec. 3.4) and now state: 'We define debt as carbon storage potential that has not been realized yet, and carbon that the atmosphere owes to vegetation'. We prefer to keep the negative sign to make clear that potential carbon and tree cover exceed carbon and tree cover in transient runs.

L315: I'm tripped up again by the use of 'typically'... Do you mean in the transient simulations? By implication in the real world? Also technically I guess this is a prediction, rather than an illustration, so many change 'show' to something less forceful.

**RESPONSE**: Yes, we mean that vegetation in transient simulations is not in equilibrium. We reworded the sentence to avoid 'typically' and in the revision we state 'Using a dynamic vegetation model, we predict that vegetation exposed to transient environmental forcing is not in equilibrium with environmental conditions ...'. In the discussion, we also provide references reporting lag effects in empirical experiments.

L320: Also change 'previous findings' to 'previous model results' or suchlike.

**RESPONSE**: We reworded as suggested to 'with our previous model results indicating that ...'.

L325: I feel like this section is missing a statement on what, if any, other studies have had to say on this topic, for Africa or elsewhere. Or indeed, is this type of analysis totally novel?

**RESPONSE**: The aim of this paragraph is to put the results presented in this study into the context of previous aDGVM studies and highlight the novelty within the 'aDGVM world'. Yet, we discuss the novel results in the context of previous studies in later sections of the discussion. We therefore prefer to keep this section focused on aDGVM. We slightly reword the section by adding references to aDGVM to make clear that it only refers to aDGVM results.

L328: This justification for why the fire runs have greater lags should perhaps come earlier in the manuscript.

**RESPONSE**: We added an explanation why fire increases lags in the results. See also our response to comments regarding L275.

L358: This needs a little more clarification. I.e. "(the strong $CO_2$ effect) can compensate for other predicted changes in climate drivers, e.g. reduced rainfall"

**RESPONSE**: We reworded as suggested: '... that the strong $CO_2$ effects can compensate for other predicted changes in climate drivers, such as reduced rainfall'.

L357: Maybe expand on what a high $CO_2$ sensitiviy might mean for the results of this paper?

**RESPONSE**: We highlighted the implications of $CO_2$ sensitivity. In particular, we added: 'If aDGVM overestimates the strength of $CO_2$ fertilization effects and the sensitivity of vegetation to elevated

CO2, the size of carbon debt due to lag effects may be overestimated, while the lag size may be underestimated. We are however confident that even with reduced CO2 sensitivity the overall response pattern would remain, although the quantities might change.'

L385: change to :prevents both savanna and forest trees recruitment"

**RESPONSE**: We reworded this statement (in response to referee 2): 'Fire rarely occurs in simulated forests, and therefore they reach equilibrium faster than other biome types. Fire activity in forests is sufficient to slightly increase times to reach equilibrium when compared to simulations with fire suppressed.'

L398: Need a reference for 'as is currently often the case'.

**RESPONSE**: We deleted 'as is currently often the case', because this statement is too bold.

L420: Would it make sense to frame the carbon debt here in terms of the overall carbon budgets of the different RCPs? For example, if emissions stabilises at the end of RCPxx, then the ecosystems would continue to keep absorbing carbon to get to their equilibrium ecosystem state and re-absorb xx PgC from the system?

**RESPONSE**: Good suggestion, we have reworded and reframed the paragraph as suggested: 'If emissions follow the RCP 2.6 scenario and stabilize after 2100, then ecosystems in Africa would continue to absorb 9 PgC or 18 PgC from the atmosphere in the presence or absence of fire to reach an equilibrium state with environmental conditions. In contrast, ecosystems would absorb 47.5 PgC or 60 PgC in the presence or absence of fire in the RCP 8.5 scenario.'

L439: I don't really understand this part. How can modifications to the initial climate conditions have an impact on the successional stage? Also, I think it's fairly standard to do pre-industrial spin-up and then transient simulations to the present day. Are you proposing an alternative approach to climate drivers here?

**RESPONSE**: We argue that initial vegetation states (rather than climate) in the model should agree with successional states of vegetation at the beginning of the transient simulation run. This can for example be achieved by considering land-use history, or by ensuring that vegetation height or age is in agreement with observation and not only variables such biomass, NPP, GPP. An approach using forest height for initialization has for example been adopted by Rödig et al. (2017) for initialization of the FORMIND model. We revised this paragraph and added the example by Rödig et al. (2017): 'Rödig et al. (2017) used the Simard et al. (2011) vegetation height product to match observed and simulated successional stages in the FORMIND model.'

Rödig, E, Cuntz, M, Heinke, J, Rammig, A, Huth, A. (2017) Spatial heterogeneity of biomass and forest structure of the Amazon rain forest: Linking remote sensing, forest modelling and field inventory. *Global Ecol Biogeogr*. 26: 1292– 1302.

L444: Further, given that we need to run transient simulations with ramping CO2, that is slightly at odds with initializing with contemporary observations.

**RESPONSE**: We don't see a contradiction here, because initialization can also consider transient simulations. For example one could use simulated and observed timeseries of LAI or NDVI for model benchmarking. Such benchmarking would require the use of transient climate and CO2 forcing.

L474: Maybe cite Julia Nabels work with TREEMIG as illustrative of the complexities of the implementation of seed dispersal?

**RESPONSE**: We cited the Nabel et al. (2013) Ecological Modelling paper: 'Nabel et al. (2013) used the TreeMig model to illustrate the complexity of simulating seed dispersal and migration in heterogeneous and variable environments.'

L500: Again, I'd argue that it 'predicts' or 'projects' or 'indicates', but perhaps not that it 'shows'.

**RESPONSE**: As suggested, we replaced 'shows' by 'indicates' and 'predicts' in the Conclusions.

**Responses to RC3 by referee #2**

General Comments

This paper considers the lag between a transient and committed vegetation state under changing CO2 and due to the disturbance effect of fire, using the aDGVM. In my view the paper is well written, clearly structured, and presents relevant and interesting results. The study is structured around 4 hypotheses which consider the current vegetation state, the impact of rate of change, the extent of change, and the effects of fire, which are novel and useful. The methods are explained clearly. The definition of equilibrium presented in equation 1 appears logical, although I wonder if there is already a published method for this that has been used in other studies. The results are presented in a logical way, and the text supports the figures throughout. I believe the conclusions are a valid interpretation of the results and that they are substantial and useful. I have some small comments on specific sections as outlined below, but otherwise I think the paper is of very good quality.

**RESPONSE**: Thank you for the positive feedback. Regarding the equilibrium conditions, please see our response below.

Specific Comments

Line 19 – Include the time period for the Devonian period to give context

**RESPONSE**: We added the period (419.2-358.9 Ma).

Line 28 - Paleocene-Eocene Thermal Maximum (PETM), a period with high carbon emissions some 56 million years ago – It would be nice to see a little more about this period and explain why the carbon emissions were high

**RESPONSE**: We modified the text and added "During the PETM, temperature increased by approximately 5-8K due to massive carbon release likely caused by volcanic activity. As temperature

increased by 6K within a 20ky period, the PETM is often considered as best analogue for current and future climate change (Zeebe 2016)".

Line 50 – is there a reference for this definition of equilibrium? I wonder if there is another method available which has been used in already published studies that can be referred to. I can see the logic of this method but some extra reference to existing methodology, and why it has been altered if necessary, would make this stronger

**RESPONSE**: We screened the ecological and land surface modeling literature, particularly the papers cited in our manuscript, but we did not find a similar mathematical definition. Essentially, our approach reflects steady state conditions as used in calculus, i.e., the first derivative has to be zero. As simulated variables are stochastic, we relax this condition by requiring that the first derivative (approximated by the difference between values of subsequent years) has to be smaller than a predefined value for a certain period.

Section 2.1 Line 110 – There aren't many PFTs represented in aDGVM. However it is mentioned in the discussion that this may cause an underestimation in lag time in forests, and as the study is focused on one savanna location I think it is enough for this study

**RESPONSE**: aDGVM only simulates two tree PFTs, savanna trees and forest trees and as mentioned by the referee and in the discussion, this prevents a detailed representation of successional dynamics and changes in the community composition. Therefore, our simulations may underestimate lag size in forests. However, despite the low number of PFTs, simulated communities and functional diversity may change in terms of population dynamics (i.e., number of trees, height and age structure), and in terms of phenology and carbon allocation patterns. These features are dynamic in aDGVM and allow simulated plants to adjust to changing environmental conditions. Note that most of our analyses were conducted for Africa at the continental scale, only in Fig 10 we used a site that is currently a savanna (but grassland or forest under low or high $CO_2$, respectively).

Line 136 – the performance of aDGVM has been evaluated in terms of vegetation, but what about fire? It would be good to see some evidence that the fire model is reliable, at least for the location picked

**RESPONSE**: In Scheiter and Higgins (2009) GCB and Scheiter et al. (2015) New Phytologist, we showed that aDGVM can reproduce broad patterns of fire activity in Africa and Australia. At local level, Governder et al. (2006) states that natural fire return intervals in Kruger National Park, South Africa, are between 4 and 5 years, and previous aDGVM simulations show that simulated return intervals are in the same order of magnitude. We did, however, not compare simulated fire activity with paleo records to assess if aDGVM can simulate fire regimes at pre-industrial or even lower $CO_2$. We added a statement: "In Scheiter and Higgins (2009) and Scheiter et al. (2015) we showed that aDGVM can simulate broad patterns of fire activity in Africa and Australia, respectively."

Section 3.6 Line 300 – can you give an explanation as to why the carbon debt continues to increase when the tree cover debt decreases?

**RESPONSE**: Both in Fig 7 and Fig 9 tree cover debt saturates because tree cover is limited by 100%, i.e., full canopy closure. As $CO_2$ increases, more and more grid cells reach closed canopy. Tree cover debt is constrained and saturates. In contrast, even if canopy closure occurs in a grid cell, biomass can further increase, for example by higher tree numbers and taller trees. Therefore, tree biomass

debt continues to increase. We added an explanation in the results, sec. 3.4: "Tree cover debt saturates decreases at higher CO2 mixing ratios because tree cover in a grid cell is constrained by canopy closure. At higher CO2 mixing ratios large fractions of Africa reach a forest state and canopy closure. Tree cover debt in these areas is zero. In contrast, biomass in a grid cells and hence biomass debt can further increase even if canopy closure occurs."

Fig 3 Bar plot – if fire is suppressed in forests (L384) would you not expect the forest results in figure 3 a and b to be the same, or would there still be some fire?

**RESPONSE**: Yes, there would still be some fire, especially in forest areas with lower tree cover. However, fire return intervals are low in these regions. We reworded: "Fire rarely occurs in simulated forests, and therefore they reach equilibrium faster than other biome types. Fire activity in forests is, however, sufficient to slightly increase times to reach equilibrium in comparison to simulations with fire suppressed."

Also from figure 3, I think it would be worth quantifying the lag time and noting in the abstract how much longer it takes to reach equilibrium per X increase in CO2, which is an important result

**RESPONSE**: Note that Fig 3 shows times required to reach an equilibrium state and not lag size. As suggested we calculated the relation between CO2 and times required to reach equilibrium, specifically we calculated averages across all biomes for simulations with fire and without fire. We added in the results and the abstract: "When averaged across all biomes and simulations with and without fire, times to reach an equilibrium state increase from approximately 242 years for 200 ppm to 898 years for 1000 ppm."

Line 256 – Lags are larger at low and intermediate CO2 mixing ratios and decrease at higher CO2. How does this fit with 'The time until vegetation reaches an equilibrium state. . .. Increase[s] with CO2' (L236)

**RESPONSE**: We do not see a contradiction in these statements as they describe different results. L236 describes times to reach equilibrium states in different biome types (Fig 3), whereas L256 describes lags between transient and equilibrium simulations (Fig 4).

Line 270 / Figure 5 and 6 – It follows that the time taken for the transient simulations to reach equilibrium is measured, but how is the time taken to reach equilibrium in equilibrium simulations measured? In other words what is the equilibrium simulation initialised from?

**RESPONSE**: It is initialized with the 'standard' method of initializing aDGVM, i. e., 100 trees with random biomass below 150 kg (uniform random distribution) and grass biomass of 0.01 kg/m^2. See also section 2.4. We added grass and tree biomasses used for initialization in the Methods section.

Line 289 – I think specifying that the debt is "larger" would be better than "higher" given the values are increasingly negative

**RESPONSE**: We reworded as suggested and replaced 'higher' by 'larger'.

Technical Comments

Line 18 – Earth's history?

**RESPONSE**: We added the s.

Line 25 – Define RCP (Representative Concentration Pathway)

**RESPONSE**: We defined RCP as suggested.

Line 91 – Does the a in aDGVM stand for anything?

**RESPONSE**: adaptive Dynamic Global Vegetation Model, it is defined in sec 2.1 and we now write it out in l. 91 as well.

Line 116 - "This approach allows to model how herbivores" – allows us to model?

**RESPONSE**: We corrected as suggested.

Line 228 – "C4 or C3-dominated vegetation if fire is present or absent" respectively.
In most of the figures C4 grassland and savanna is labeled, but woodland and forest is not labeled as C3 despite being referred to in the text as C3

**RESPONSE**: We use two different notations: the different biome types (e.g. C3 grassland, C4 grassland, forest) and C3/C4 dominated biomes where we aggregate several biomes. We do not label woodland and forest explicitly as C3 woodland or C3 forest, because these biomes are always dominated by C3 trees; a C4 dominated forest state is not possible. We checked the manuscript to ensure that there is no ambiguity and we modified the biome classification section to define C3 and C4 dominated biomes.

[revised manuscript text omitted]